# Osteocyte CIITA aggravates osteolytic bone lesions in myeloma

Huan Liu[1,2,3], Jin He[1,2], Rozita Bagheri-Yarmand [4], Zongwei Li[1,2], Rui Liu[3], Zhiming Wang[1,2], Duc-hiep Bach[1], Yung-hsing Huang[1], Pei Lin[5], Theresa A. Guise[4], Robert F. Gagel[4 ✉] & Jing Yang [1,2 ✉]

Osteolytic destruction is a hallmark of multiple myeloma, resulting from activation of osteoclast-mediated bone resorption and reduction of osteoblast-mediated bone formation. However, the molecular mechanisms underlying the differentiation and activity of osteoclasts and osteoblasts within a myelomatous microenvironment remain unclear. Here, we demonstrate that the osteocyte-expressed major histocompatibility complex class II transactivator (CIITA) contributes to myeloma-induced bone lesions. CIITA upregulates the secretion of osteolytic cytokines from osteocytes through acetylation at histone 3 lysine 14 in the promoter of TNFSF11 (encoding RANKL) and SOST (encoding sclerostin), leading to enhanced osteoclastogenesis and decreased osteoblastogenesis. In turn, myeloma cell–secreted 2-deoxy-D-ribose, the product of thymidine catalyzed by the function of thymidine phosphorylase, upregulates CIITA expression in osteocytes through the STAT1/IRF1 signaling pathway. Our work thus broadens the understanding of myeloma-induced osteolysis and indicates a potential strategy for disrupting tumor-osteocyte interaction to prevent or treat patients with myeloma bone disease.

[1] Houston Methodist Cancer Center, Houston Methodist Research Institute, Houston Methodist Hospital, Houston, TX, USA. [2] Department of Lymphoma and Myeloma, Division of Cancer Medicine, The University of Texas MD Anderson Cancer Center, Houston, TX, USA. [3] Cancer Research Center, School of Medicine, Xiamen University, Xiamen, China. [4] Department of Endocrine Neoplasia and Hormonal Disorders, Division of Internal Medicine, The University of Texas MD Anderson Cancer Center, Houston, TX, USA. [5] Department of Hematopathology, Division of Pathology and Laboratory Medicine, The University of Texas MD Anderson Cancer Center, Houston, TX, USA. ✉email: rfgagel@gmail.com; jyang2@houstonmethodist.org

Osteocytes are the most abundant cell type in bone tissue and are generally considered terminally differentiated cells of the osteoblast lineage[1]. They are the central regulator of bone remodeling through their bone-resorbing osteoclasts and bone-forming osteoblasts[2]. Despite being buried within mineralized bone matrix, osteocytes can modulate hematopoietic monocytic precursors developing into mature osteoclasts as well as regulate the differentiation of precursor mesenchymal stem cells (MSCs) into osteoblasts through communication with cells on the bone surface and those within bone marrow. Large amounts of soluble cytokines are secreted from osteocytes[2]. For example, osteocytes are the primary secretors of sclerostin, a small glycoprotein encoded by the *Sost* gene[3]. As a requisite for osteoblast development and activity, Sclerostin binds to the Wnt co-receptors lipoprotein receptor-related proteins 4/5/6 thereby inhibiting the Wnt/β-catenin signaling pathway[3]. Osteocytes are also the main source of receptor activator of nuclear factor κB ligand (RANKL), which stimulates the generation and activity of osteoclasts[4]. Recently, osteocytes were shown to play an important role in the development and progression of bone disease in some tumors, including multiple myeloma[5,6].

Multiple myeloma is a B cell malignancy characterized by clonal expansion of malignant plasma cells within bone marrow. More than 80% of patients with myeloma develop lytic lesions in bone, thereby causing pathologic fractures, severe bone pain, spinal cord compression, and hypercalcemia[7]. Myeloma-induced bone destruction results from increased osteoclast-mediated bone resorption and decreased osteoblast-mediated bone formation;[8] and osteocytes are a critical driver in this process. For example, knockout of sclerostin expression in osteocytes or pharmaceutical inhibition of sclerostin can reduce bone lesions in myeloma mouse models[9], indicating that the interaction between tumor cells and osteocytes is an essential pathway in the genesis of lytic lesions seen in myeloma patients. However, the molecular mechanism by which osteocytes express and secrete osteolytic cytokines, stimulated by myeloma cells, still needs further exploration.

Here, through a combination of in vitro and in vivo studies and the use of patient samples, we demonstrate that the histone acetyltransferase major histocompatibility complex (MHC) class II transactivator (CIITA), expressed in osteocytes, contributes to myeloma-induced bone lesions. CIITA is a co-activator that regulates interferon γ–induced transcription of MHC class II genes, ensuring intracellular transport and surface presentation of antigenic peptides[10]. Histones H3 and H4 are acetylated at the promoter of MHC class II genes, which depends on the level of CIITA expression and binding activity[11]. In addition to its function in adaptive immunity, CIITA is an activator of osteoclast differentiation and bone homeostasis[12], since globally overexpressing CIITA in mice increases RANKL-mediated signaling and causes bone resorption and severe spontaneous osteoporosis. However, the biologic function and mechanism of CIITA in osteocyte-mediated bone homeostasis, especially in myeloma bone disease, remain unclear. Here, we demonstrate that CIITA upregulates the expression of osteolytic cytokines RANKL and sclerostin through histone acetylation at H3K14 in the promoters of *TNFSF11* and *SOST* genes, leading to decreased osteoblast differentiation, enhanced osteoclastogenesis, and bone lesions. Our research thus defines CIITA as a molecular link bridging osteocytes and myeloma bone disease.

## Results

**Osteocytes contributes to myeloma bone lesions in mice**. To examine the functional role of osteocytes in myeloma-induced bone destruction, we first intrafemorally injected murine myeloma cells Vk12598 into C57BL/6 mice (Fig. 1a). Mice that did not receive myeloma cells served as controls. After 6 weeks, we measured the establishment of myeloma by detecting increased levels of circulating M-proteins, an indicator of myeloma burden, and observed the development of lytic lesions in mouse femurs (Fig. 1b). We isolated and characterized primary osteocytes from the mouse femurs (Supplementary Fig. 1). We cultured those osteocytes to generate conditioned medium (CM). Cultures of MSCs with the CM of osteocytes isolated from tumor-bearing mice had less Alizarin red-S staining, which reflects osteoblastogenesis, than cultures of the CM from osteocytes of control mice (Fig. 1c). In the in vitro osteoclastogenesis assay, the CM of osteocytes isolated from tumor-bearing mice, compared to that of control mice, induced higher numbers of multinucleated tartrate-resistant acid phosphatase–positive (TRAP$^+$) mature osteoclasts (Fig. 1d). Furthermore, quantitative polymerase chain reaction (qPCR) analysis showed significant upregulation of several osteocyte-specific cytokines[13] in the osteocytes isolated from myeloma-bearing mice, compared to those obtained from control mice (Fig. 1e). These cytokines included tumor necrosis factor ligand superfamily member 11 (*Tnfsf11*), the gene encoding RANKL, which can induce osteoclast differentiation, and *Sost*, the gene encoding sclerostin, which can inhibit osteoblast formation and function. These results indicate that the osteocytes from myeloma-bearing mice with lytic lesions enhance the activation of osteoclastogenesis and suppress osteoblastogenesis.

**Osteocyte CIITA mediates myeloma-induced bone lesions**. We next examined the alteration of gene expression in osteocytes isolated from Vk12598 myeloma–bearing mice, compared to those in control mice that did not receive myeloma cells, using whole-transcriptome RNA sequencing (RNA-seq) (Fig. 1a). Computational overlapping of genes with the Molecular Signatures Database (Broad Institute) hallmark gene sets[14,15] suggested significant enrichment of genes in mineral absorption signaling (Fig. 1f), confirming the functional role of osteocytes in bone. Paired differential analysis identified 49 downregulated genes and 19 upregulated genes in the osteocytes of myeloma-bearing mice compared to those in osteocytes from control mice (Fig. 1g). Unsupervised hierarchical clustering of osteocytes using RNA-seq gene expression data showed a response-specific gene expression signature, including four candidates: *Clec9a*, *Riken*, *Adgrg5*, and *Ciita* that were upregulated in the osteocytes from myeloma-bearing mice (Fig. 1g, h). Consistent with the RNA-seq results, we confirmed that osteocytes isolated from an additional five myeloma-bearing mice had increased expression of *Ciita* protein (Fig. 1i) and mRNA levels of four candidates (Fig. 1j and Supplementary Fig. 2a), compared to expression in control osteocytes.

We next determined the relationship of gene expression levels in osteocytes with the extent of myeloma-induced osteolytic lesions in mice. We found a robust negative correlation of osteocyte *Ciita* expression levels with the percentage of bone volume/total volume (BV/TV) (Fig. 1k) and the extent of bone formation, including mineral apposition rate (MAR) and % bone surface undergoing mineralization during the interlabel time (BFR/BS, Fig. 1l), percentages of osteoid surface (OS/BS) (Fig. 1m, left), and the percentage of the bone surface lined with osteoblasts (Ob.S/BS) (Fig. 1m, right) in mouse femurs, as well as levels of mouse serum procollagen type I N-terminal propeptide (PINP), a bone formation marker (Fig. 1n, left). In contrast, we observed a strong positive correlation between the level of osteocyte *Ciita* expression and the extent of bone resorption, including the level of C-telopeptide of type I collagen (CTX-1) in mouse serum (Fig. 1n, right), percentage of bone surface eroded by osteoclasts (ES/BS) (Fig. 1o, left), and percentage of bone surface covered with osteoclasts (Oc.S/BS) (Fig. 1o, right). There

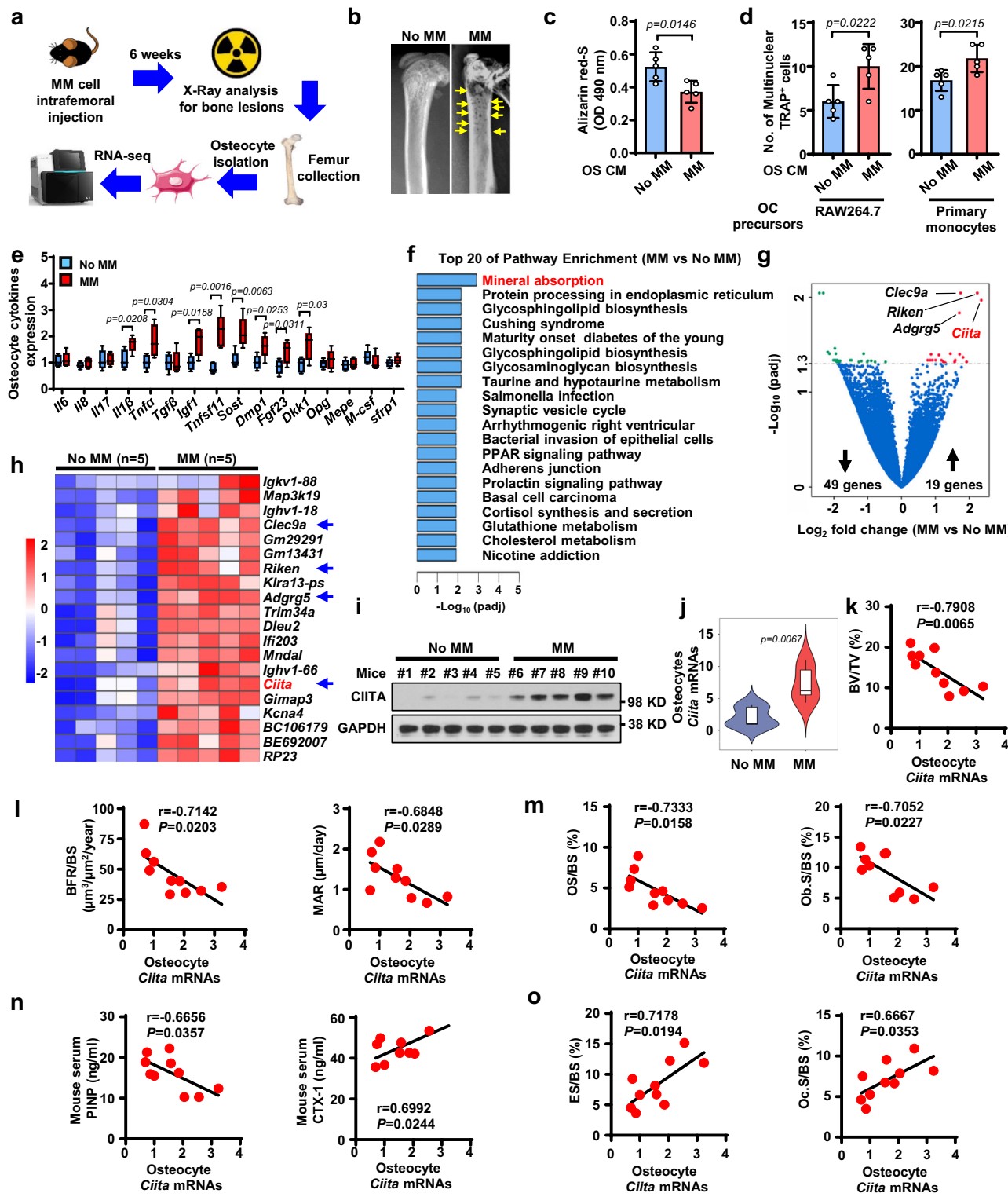

were no obvious correlations between mRNA levels of *Clec9a*, *Riken*, or *Adgrg5* in osteocytes and the bone intensity in myeloma-bearing mice (Supplementary Fig. 2b). Our results indicate that osteocyte-expressed CIITA is associated with the pathogenesis of myeloma-induced bone destruction.

To examine the function of osteocyte CIITA in bone in the presence of myeloma, we generated osteocyte-specific conditional *Ciita*-knockout mice using a Flp-Cre strategy. In the knockout

mice, CIITA was not expressed in osteocytes but was expressed in other bone cells, such as MSCs and osteoblasts (Fig. 2a). To establish myeloma in mice, we intrafemorally injected Vk12598 cells into wild-type and *Ciita*-knockout mice. Mice that did not receive myeloma cells served as controls for baseline level of bone density. Six weeks after myeloma cell injection, we found increased levels of M-proteins in both the knockout mice and the wild-type mice (Fig. 2b), indicating establishment of

**Fig. 1 Transcriptome profile of osteocytes isolated from myeloma-bearing mice identifies CIITA as a potential promoter for myeloma-induced bone destruction. a** Schematic workflow of RNA-seq of osteocytes isolated from myeloma-bearing mice. **b** Representative x-rays of femurs from control mice (No MM) and Vk12598 cell–injected mice (MM) at week 6. Yellow arrows, lytic lesions. **c–e** The CM of osteocytes (OS CM) was obtained from cultures of osteocytes isolated from control mice or MM mice ($n = 5$/group). Shown are representative summarized data of Alizarin red-S staining in cultures of MC3T3-E1 cells with OS CM (**c**), numbers of multinuclear ($\geq 3$) TRAP$^+$ cells in cultures of Raw264.7 (**d**, left) or primary monocytes (**d**, right) with OS CM, and the expression profile of cytokine genes in osteocytes (**e**). Data are presented as mean ± SD. Box plot (**e**) indicates median, 25th and 75th percentiles, and minima and maxima of the distribution. **f** Pathway enrichment analysis in the osteocytes from myeloma-bearing mice compared to those from control mice. **g** Volcano plot showing genes with a cutoff fold-change of $\geq 2$ or $\leq -2$ and a $P$ value of $<0.05$. $P$ values in (**f–g**) were calculated using the DEseq2 package (two-sided, no correction for multiple testing). **h** Heat-map showing the expression profile of upregulated genes in the osteocytes of MM mice compared to those of control mice ($n = 5$/group). **i–j** Protein (**i**) and mRNA (**j**) levels of CIITA in the osteocytes isolated from mice. **i** Shown are representative of two independent experiments. Violin plot (**j**) indicates median, 25th and 75th percentiles, and minima and maxima of the distribution. Data are presented as mean ± SD. **k–o** Correlation coefficients between the levels of *Ciita* mRNAs in osteocytes with percentages of BV/TV (**k**), levels of MAR and BFR/BS (**l**), OS/BS and Ob.S/BS (**m**), serum PINP and CTX-1 (**n**), and percentages of ES/BS and Oc.S/BS (**o**) in myeloma-bearing mice compared to control mice ($n = 10$/group). $P$ values were calculated by unpaired two-tailed $t$ test (**c–e**, **j**). The correlations were evaluated using Pearson coefficient with two-tailed $P$ value. r, correlation coefficient (**k–o**).

myeloma. As shown in Fig. 2c, myeloma-bearing wild-type mice displayed severely reduced trabecular bone in the marrow, while knockout of CIITA in osteocytes significantly reversed such effects. Quantitative μCT data showed a higher level of BV/TV, average trabecuar thickness (Tb.Th), trabecular bone density (Tb.N), and a lower level of average distance between trabecular bone (Tb.Sp) in the myeloma-bearing *Ciita*-knockout mice than in the wild-type mice (Fig. 2d, g). There was no significant change in marrow trabecular bones between the wild-type and knockout mice without myeloma (Fig. 2c–g). Bone histomorphometric analysis showed lower levels of ES/BS and Oc.S/BS and higher levels of OS/BS and Ob.S/BS in the *Ciita*-knockout mice than in wild-type mice when myeloma was established (Fig. 2i–k). In agreement with these findings, the bone formation rate was increased in the myeloma-bearing *Ciita*-knockout mice compared to wild-type mice (Fig. 2l).

To examine the role of osteocyte CIITA in osteoblast and osteoclast differentiation in myeloma in vitro, we isolated primary osteocytes from the femurs of wild-type or *Ciita*-knockout mice and generated osteocyte CM. Culturing MSCs or osteoclast precursors with CM derived from wild-type osteocytes isolated from myeloma-bearing mice showed less Alizarin red-S staining (Fig. 2m) and more multinucleated TRAP$^+$ osteoclast formation (Fig. 2n, left) than observed in myeloma-bearing *Ciita*-knockout mice. Similarly, we have observed increased multinucleated TRAP$^+$ cells in the cultures of osteoclast precursor monocytes cultured in the CM of myeloma cell-exposed primary osteocytes with knocked down *Ciita* (Fig. 2n, right). However, this effect was not seen in cultures with CM of osteocytes from mice without myeloma (Fig. 2m, n). We also observed lower levels of *Tnfsf11* and *Sost* mRNAs, as well as lower levels of RANKL and sclerostin proteins, in *Ciita*-knockout osteocytes compared to wild-type osteocytes from mice bearing myeloma, while there was no difference in the expression of those cytokines in osteocytes from mice without myeloma (Fig. 2o, p).

We also knocked down CIITA expression in murine osteocyte cell lines MLO-Y4 and MLO-A5 using murine *Ciita* short hairpin RNA (shRNA), and cells with non-target shRNAs (sh*Ctrl*) served as controls (Supplementary Fig. 3a). There was no difference in apoptosis or viability between control and knockdown cells (Supplementary Fig. 3b, c), indicating that the modification of CIITA does not affect osteocyte growth or survival. We then cultured control or *Ciita*-knockdown osteocytes with myeloma ARP-1 cells and collected the osteocyte CM. To examine osteoclast and osteoblast differentiation, we cultured the osteoblast precursor MC3T3-E1 and the osteoclast precursor Raw264.7 with the osteocyte CM. We found that cultures with the CM of *Ciita*-knockdown osteocytes had less ability to reduce osteoblast

formation (Supplementary Fig. 3d) and enhance osteoclast differentiation (Supplementary Fig. 3e, left) than did cultures with the CM of control osteocytes. We then collected CM from cultures of myeloma cells and wild type or *Ciita* knocked down primary osteocytes. We found that primary monocytes culturing in the CM of *Ciita* knockdown osteocytes showed reduced numbers of TRAP$^+$ cells when compared with those from wild type (Supplementary Fig. 3e, right). Together, these findings indicate that CIITA expressed by osteocytes mediates myeloma-induced bone lesions.

**The TP–CIITA axis enhances cytokine secretion from osteocytes.** Myeloma cells highly express thymidine phosphorylase (TP) and secrete 2-deoxy-D-ribose (2DDR), the product of thymidine catalyzed by TP function[7]. Myeloma cell TP/2DDR has been shown to play an important role in the pathogenesis of myeloma bone disease[7]. We found that the levels of RANKL and sclerostin in the serum of mice injected with high TP–expressing myeloma cells ARP-1 and RPMI8226 were much higher than those in mice injected with low TP–expressing MM.1 S myeloma cells (Supplementary Fig. 4a). Knockdown of TP in ARP-1 cells reduced cytokine levels, while overexpression of TP in MM.1 S cells enhanced cytokine levels in mouse serum (Supplementary Fig. 4b). Treatment of TP inhibitors 7DX or TPI reduced circulating cytokine levels in mice injected with ARP-1 or RPMI8226 cells (Supplementary Fig. 4c). In vitro studies using osteocyte cell lines MLO-Y4 and MLO-A5 found that co-culture with sh*Ctrl* ARP-1 cells upregulated the osteocytes' mRNA levels of *Tnfsf11* and *Sost* than those cultured alone, while this effect was reduced in osteocytes co-cultured with sh*TP* ARP-1 cells (Supplementary Fig. 4d, e). Addition of 2DDR to the osteocytes significantly increased the expression of several cytokines, including *Tnfsf11* and *Sost*, which are secreted from osteocytes (Supplementary Fig. 4f). We thus hypothesized that myeloma cell TP/2DDR upregulate osteocyte cytokine expression through stimulation of CIITA expression in osteocytes. To test our hypothesis, we co-cultured myeloma cells and MLO-Y4 or MLO-A5 osteocytes. We found that co-culture with the ARP-1 cells (high TP expression) induced higher levels of CIITA mRNA (Fig. 3a) and protein (Fig. 3b) in osteocytes than did co-culture with MM.1 S cells (low TP expression). Knockdown of TP in ARP-1 cells reduced CIITA expression in MLO-Y4 or MLO-A5 cells, while overexpression of TP in MM.1 S enhanced CIITA expression (Fig. 3a, b). Correspondingly, addition of 2DDR increased levels of CIITA mRNA (Fig. 3c) and protein (Fig. 3d) in MLO-Y4 and MLO-A5 cells. Since the CIITA transcripts in immune cells have been shown to be driven by promoters I, III, and IV, we wanted to determine the isoform(s) by which 2DDR

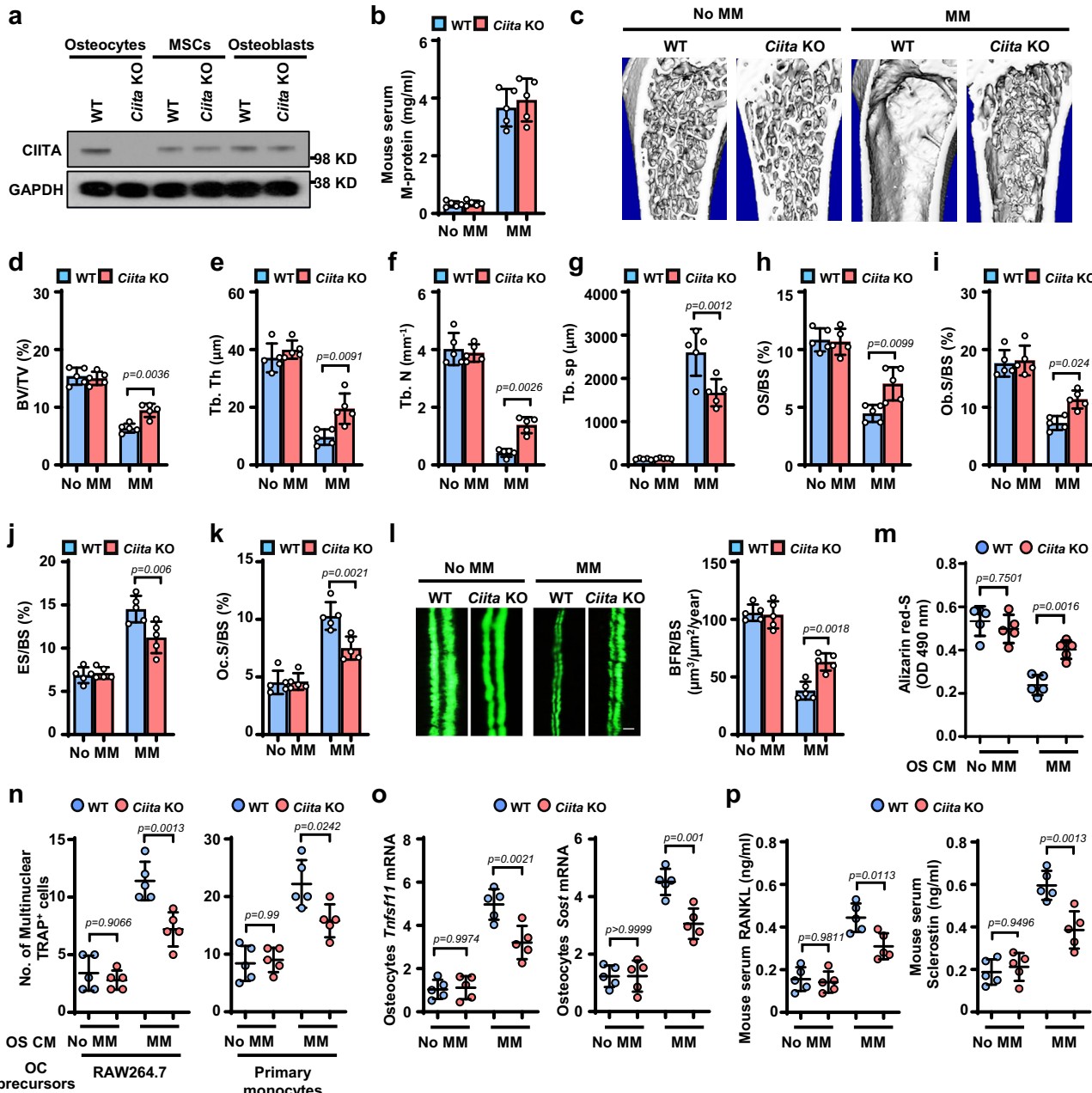

**Fig. 2 Deficiency of CIITA expression in osteocytes impairs myeloma-induced bone lesions. a** Western blots show the expression of CIITA in osteocytes, MSCs, and osteoblasts isolated from wild-type (WT) or osteocyte-specific *Ciita*-knockout (*Ciita* KO) mice. The expression of GAPDH served as protein loading controls. Images shown are representative of two independent experiments. **b–k** WT and osteocyte-specific *Ciita* KO mice were intrafemorally injected with the murine myeloma cell line Vk12598 (1 × 10^6 cells/mouse). The mice not receiving myeloma cells (No MM) served as controls. Shown are the concentrations of M-proteins in mouse sera (**b**), representative μ-CT images of mouse femurs (**c**), and percentages of BV/TV (**d**), Tb.Th (**e**), Tb.N (**f**), Tb.Sp (**g**), OS/BS (**h**), Ob.S/BS (**i**), ES/BS (**j**), and Oc.S/BS (**k**). **l** BFR/BS was measured by calcein injection, and the bone sections were imaged and analyzed. Shown are representative images and summarized data of bone formation in mouse femurs. Scale bar: 20 μm. **m-p** Osteocytes isolated from WT or *Ciita*-KO mice injected with or without myeloma cells, were cultured for generating the osteocyte CM. The osteoblast precursor MC3T3-E1 and the osteoclast precursor Raw264.7 were cultured with the osteocyte CM. After culturing, the cells were stained with Alizarin red-S for osteoblast differentiation or TRAP staining for osteoclast differentiation. Shown are the summarized data from Alizarin red-S staining (**m**) and the numbers of multinuclear (≥3) TRAP+ cells (**n**, left). Real-time PCR and ELISA assays show the relative expression of *Tnfsf11* and *Sost* mRNA in osteocytes (**o**) and the serum levels of RANKL or sclerostin in mice (**p**). (**n**, right) Primary osteocytes isolated from mouse femurs were knocked down by *Ciita* siRNA (si*Ciita*). The CM of WT or si*Ciita* was cultured with myeloma cells for 48 hours. Data are mean ± SD (*n* = 5 mice/group). *P* values were determined using one-way ANOVA with Tukey's multiple comparisons test.

regulates CIITA expression in osteocytes. By examining two murine osteocyte cell lines and primary osteocytes isolated from mice, we found that the isoform IV is the dominant form in osteocytes and addition of 2DDR enhances *CIITA* IV mRNA in primary osteocytes (Supplementary Fig. 5).

We next examined the signaling pathways by which 2DDR may regulate CIITA expression in osteocytes. We observed that the levels of phosphorylated Syk and STAT1 were upregulated in osteocytes cultured with 2DDR, while the levels of phosphorylated ERK1/2, Akt, and JAK1, the non-phosphorylated kinases,

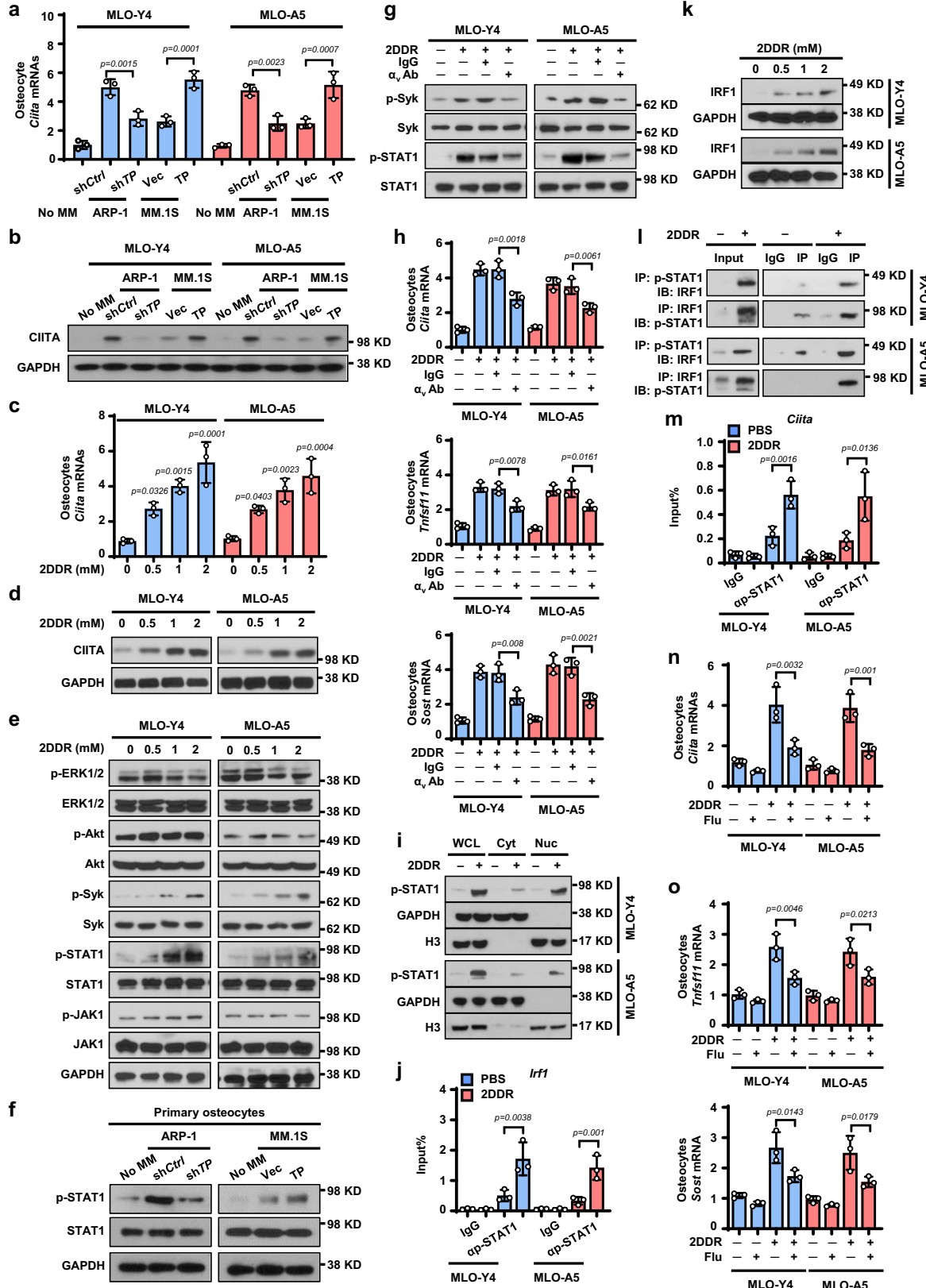

and GAPDH as a protein loading control were not changed (Fig. 3e). We observed a similar trend in primary osteocytes cultured with myeloma cells that had high expression levels of TP (sh*Ctrl* ARP-1 or *TP* MM.1 S cells, Fig. 3f). Osteocytes express the integrin $\alpha_v\beta_3$, the receptor of 2DDR[7,16]. We found that addition of the antibody against $\alpha_v$, but not addition of IgG controls,

significantly reduced 2DDR effects on Syk, STAT1 phosphorylation, *Ciita, Tnfsf11*, and *Sost* expression in MLO-Y4 and MLO-A5 cells (Fig. 3g, h). We separated the cytosol and nuclear fractions of MLO-Y4 or MLO-A5 cells and observed that phosphorylated STAT1 was localized mainly in the nuclear fraction of 2DDR-incubated osteocytes (Fig. 3i). Chromatin immunoprecipitation

**Fig. 3 Myeloma TP/2DDR enhances CIITA expression in osteocytes via the STAT1/IRF1 signaling pathway. a–b** The levels of *CIITA* mRNA (**a**, n = 3 biological replicates) or protein (**b**) in MLO-Y4 or MLO-A5 osteocytes co-cultured with the high TP–expressing ARP-1 cells carrying non-targeted control shRNAs (sh*Ctrl*) or TP shRNAs (sh*TP*) or co-cultured with low TP–expressing MM.1 S cells carrying empty control vector (*Vec*) or TP cDNAs (*TP*). **c–e** Levels of *CIITA* mRNA (**c**, n = 3 biological replicates) and protein (**d**) as well as phosphorylated (p) or non-phosphorylated ERK1/2, Akt, Syk, STAT1, and JAK1 (**e**) in MLO-Y4 and MLO-A5 cells cultured with 2DDR. (GAPDH: protein loading controls). **f** Levels of the p-STAT1 or STAT1 in primary osteocytes cultured with the CM of sh*Ctrl* or sh*TP* ARP-1 cells or with the CM of *Vec* or *TP* MM.1 S cells. **g–h** Levels of p-SyK, Syk, p-STAT1, or STAT1 (**g**), *Ciita*, *Tfsf11*, and *Sost* mRNAs (**h**, n = 3 biological replicates) in MLO-Y4 or MLO-A5 cells cultured with 2DDR, IgG control, or anti–α$_v$ antibody (Ab). **i** Levels of p-STAT1 in whole-cell lysates or cytosolic or nuclear fractions of MLO-Y4 and MLO-A5 cells cultured without or with 2DDR. Loading controls: GAPDH for cytosol fractions (Cyt); H3 for nuclear fractions (Nuc). WCL, whole cell lysate. **j** The enrichment of p-STAT1 in the promoter of *Irf1* gene in MLO-Y4 and MLO-A5 cells cultured with 2DDR. IgG levels served as controls. **k** The protein levels of IRF1 in MLO-Y4 and MLO-A5 cells cultured with 2DDR. **l** Co-immunoprecipitation of p-STAT1 with IRF1 in MLO-Y4 and MLO-A5 cells cultured with 2DDR. **m** The enrichment of p-STAT1 in the promoter of *Ciita* gene in MLO-Y4 and MLO-A5 cells cultured with 2DDR (n = 3 biological replicates). **n–o** The relative expression of *Ciita* (**n**), *Tnfsf11* and *Sost* (**o**) mRNA in MLO-Y4 and MLO-A5 cells cultured with 2DDR in the absence or presence of STAT1 inhibitor fludarabine (Flu; 10 μM) (n = 3 biological replicates). **b**, **d**, **e–g**, **i**, **k**, and **l** are representative of two independent experiments. Data are mean ± SD. *P* values were determined using one-way ANOVA with Tukey's multiple comparisons test.

(ChIP) assays showed that phosphorylated STAT1 bound to the promoter of the transcription factor interferon regulatory factor 1 (*Irf1*) in 2DDR-incubated osteocytes (Fig. 3j). Culture with 2DDR enhanced IRF1 expression in osteocytes (Fig. 3k). Furthermore, we immunoprecipitated osteocyte lysates with an anti–phosphorylated STAT1 antibody and observed the interaction between phosphorylated STAT1 and IRF1 (Fig. 3l). The ChIP assay showed the enrichment of phosphorylated STAT1 in *Ciita* promoter in 2DDR-incubated osteocytes (Fig. 3m). Addition of the STAT1 inhibitor fludarabine significantly reduced the mRNA levels of *Ciita* (Fig. 3n), *Tnfsf11*, and *Sost* in osteocytes cultured with 2DDR (Fig. 3o). These results indicate that myeloma cell TP/2DDR enhances CIITA expression in osteocytes via the STAT1/IRF1 signaling pathway.

To examine the effects of CIITA on myeloma cell TP/2DDR–induced cytokine expression in osteocytes, we used shRNAs to knock down CIITA in MLO-Y4 and MLO-A5 cells and found that the knockdown significantly reduced the effect of 2DDR on *Sost* mRNA levels in osteocytes (Fig. 4a). We pretreated sh*Ctrl* or sh*Ciita* osteocytes with 2DDR, washed out the 2DDR, and collected the osteocyte CM. We found that culture of the osteoblast precursors MC3T3-E1 with the sh*Ciita* osteocyte CM had greater ability to induce osteoblast differentiation (Fig. 4b) and expression of osteoblast differentiation–associated genes *alkaline phosphatase* (*Alp*), *Collagen Type I Alpha 1* (*Col1a1*), and *gamma-carboxyglutamic acid-containing protein* (*Bglap*) (Fig. 4c) compared to cultures with the CM of sh*Ctrl* osteocytes. Correspondingly, CIITA knockdown in osteocytes reduced the effects of 2DDR on *Tnfsf11* mRNA levels (Fig. 4d). Culture of osteoclast precursors Raw264.7 or primary monocytes with the CM of *Ciita* knockdown osteocytes had decreased ability to induce osteoclast differentiation (Fig. 4e) and expression of osteoclast differentiation marker genes *Trap*, *calcitonin receptor* (*Calcr*), and *Cathepsin K* (*Ctsk*) compared to culture with the CM of sh*Ctrl* osteocytes (Fig. 4f). In addition, we cultured osteocytes isolated from wild-type or *Ciita*-knockout mice with 2DDR to generate the osteocyte CM, and we found similar results in osteocyte-mediated osteoblastogenesis, osteoclastogenesis, and cytokine production (Fig. 4g-j). Similar results were obtained when we cultured primary monocytes with myeloma-exposed primary osteocytes with knocked down *Ciita* (Fig. 4j, right). These findings demonstrate that myeloma cell TP/2DDR upregulates CIITA expression in osteocytes and that the increased CIITA enhances osteocyte expression of osteolytic cytokines, leading to disruption of the balance between osteoclastogenesis and osteoblastogenesis.

**CIITA boosts histone acetylation in cytokine gene promoter.** Histone acetylation is a key epigenetic modification that influences tissue and context-specific gene expression and is associated with gene activation[17]. CIITA is a histone acetyltransferase that regulates acetylation of histones H3 and H4 at the MHC class II promoter[11]. However, its function and mechanism in regulation of cytokines remain unclear. With addition of 2DDR, we found higher levels of total histone H3 acetylation, but not H4 acetylation, in osteocytes compared to those in osteocytes without 2DDR (Fig. 5a). Addition of 2DDR also upregulated levels of acetylation at histone 3 lysine 14 (H3K14) but did not change the levels of acetylation at histone 3 lysine 27 (H3K27) (Fig. 5b). Knockdown of CIITA in osteocytes reversed the effect of 2DDR on the accumulation of H3K14 acetylation (Fig. 5c). ChIP PCR assay showed enrichment of H3K14 acetylation and CIITA proteins in the promoter of *Tnfsf11* or *Sost* in osteocytes cultured with 2DDR (Fig. 5d), while knockdown of CIITA in osteocytes reduced such enrichment (Fig. 5e). To determine the impact of CIITA's HAT activity on osteocyte cytokine production, we transfected primary human osteocytes with wild-type CIITA or CIITAΔAD, a dominant-negative mutant that lacks the active domain at amino acids 1–160 in CIITA. We found that the wild-type CIITA enhanced the expression of *Tnfsf11* or *Sost* in osteocytes, while the mutation had a reduced effect (Fig. 5f). These findings reinforce the role of CIITA in regulation of myeloma 2DDR–induced H3K14 acetylation in the promoter of *Tnfsf11* or *Sost*, leading to the enhanced cytokine gene transcription in osteocytes.

Because CIITA is a non-DNA binding transcriptional coactivator, we sought the link that bridges CIITA protein to specific DNA regions in the cytokine promoter. Using coimmunoprecipitation of an antibody against CIITA and analysis of the antibody's pull-down proteins with mass spectrometry, we identified a candidate: the transcriptional factor AP2α (Fig. 6a and Supplementary Table 1). To determine the interaction of CIITA and AP2α, we mixed the lysate of HEK293T cells overexpressing c-myc–tagged CIITA (*c-myc-CIITA*) with the lysate of AP2α-expressing HEK293T cells and pulled down proteins with an anti–c-myc antibody. We detected either CIITA or AP2α proteins in the lysates (Fig. 6b). This interaction was also obtained in *c-myc-CIITA* osteocytes (Fig. 6c). To examine the endogenous interaction of CIITA and AP2α in osteocytes, we cultured MLO-Y4 and MLO-A5 osteocytes with or without 2DDR and found that CIITA coprecipitated with AP2α (Fig. 6d).

To determine whether AP2α binds to *Tnfsf11* or *Sost* promoter, we examined a 1.3-kilobase region around the *Tnfsf11* transcriptional start site and a 1.5-kilobase region around the *Sost* transcriptional start site. According to the promoter sequence, we predicted one AP2α binding site (−800 base pairs [bp]) in the *Tnfsf11* promoter (Fig. 6e) and three AP2α binding sites (−840 bp,

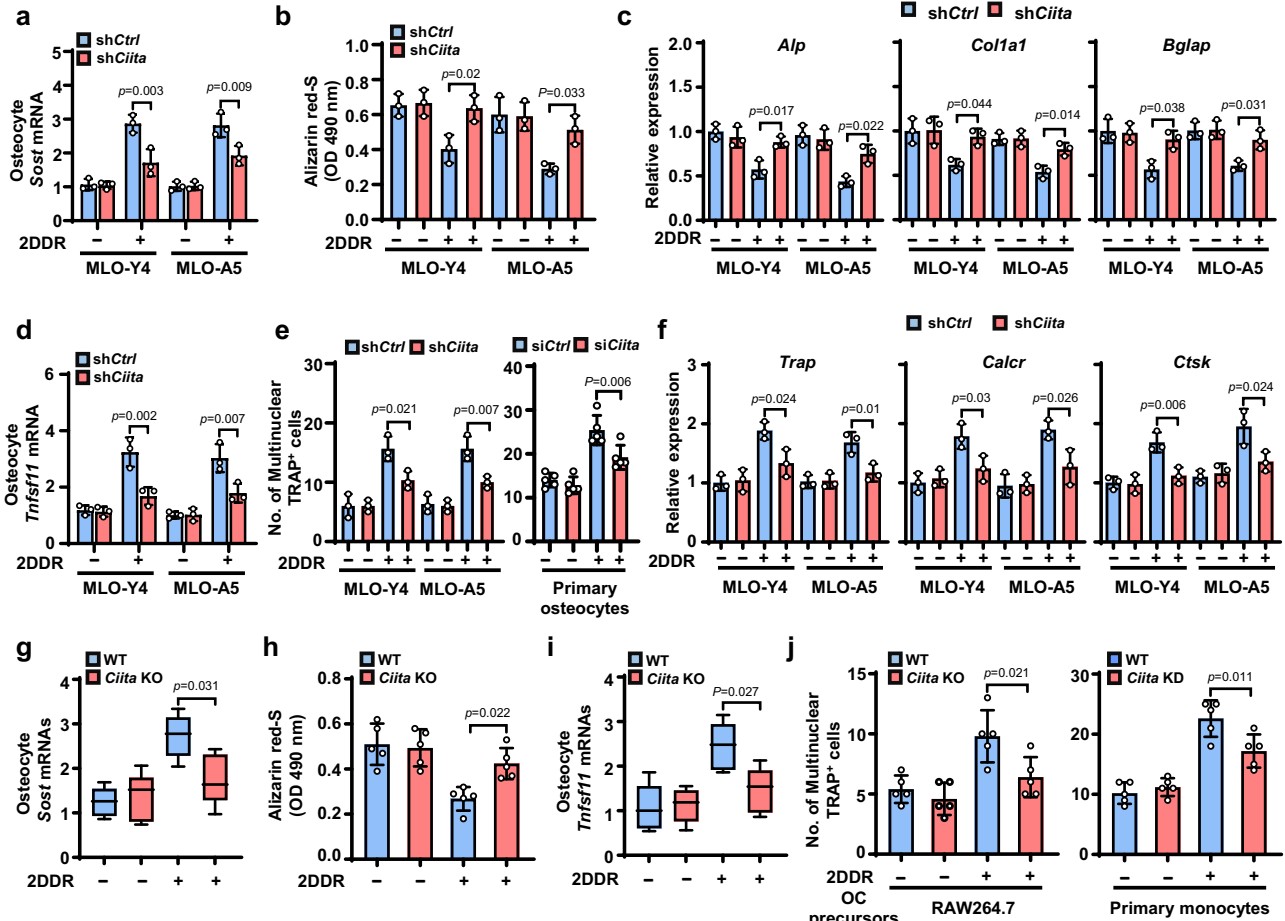

**Fig. 4 Depletion of CIITA in osteocytes abrogates effects of 2DDR on cytokine secretion, osteoclastogenesis, and suppressed osteoblastogenesis.** MLO-Y4 and MLO-A5 osteocytes carried with non-targeted control shRNAs (shCtrl) or Ciita shRNAs (shCiita) were cultured with or without 2DDR for 2 days, given fresh medium, and cultured for another 2 days, and then the cells and the CM were collected. The osteoblast precursor MC3T3-E1 and the osteoclast precursors Raw264.7 cells or primary monocytes were cultured with or without the osteocyte CM. After culturing, the cells were subjected to Alizarin red-S staining to assess osteoblast differentiation, to TRAP staining to assess osteoclast differentiation, or to real-time PCR to assess the differentiation-associated gene expression. **a–f** Shown are summarized data for Sost mRNAs (**a**), Alizarin red-S staining (**b**), the relative expression of osteoblast differentiation–associated genes Alp, Col1a1, and Bglap (**c**), Tnfsf11 mRNA (**d**), the numbers of multinuclear (≥3) TRAP+ cells (**e**), and the relative expression of osteoclast differentiation–associated genes Trap, Calcr, and Ctsk (**f**). Data are mean ± SD (n = 3 biological replicates). **g–j** Osteocytes isolated from WT or osteocyte-specific Ciita-KO mice (n = 5 mice/group) were cultured with or without 1 mM 2DDR for 2 days for collecting the cells and the osteocyte CM. MC3T3-E1 or Raw264.7 cells cultured with osteocyte CM were subjected to Alizarin red-S staining, TRAP staining, or real-time PCR. Shown are summarized data for Sost mRNA levels in osteocytes (**g**), Alizarin red-S staining (**h**), Tnfsf11 mRNA levels in osteocytes (**i**), and numbers of multinuclear (≥3) TRAP+ cells (**j**, left). Box plots (**g**, **i**) indicate median, 25th and 75th percentiles, and minima and maxima of the distribution. Data are presented as mean ± SD. (**j**, right) Primary osteocytes wild-type or carried with Ciita siRNAs (siCiita) were cultured in the same fashion as in (**a**) to collect the CM. The numbers of multinuclear (≥3) TRAP+ cells were counted in the cultures of primary monocytes with or without the osteocyte CM. Data are means ± SD. P values were determined using one-way ANOVA with Tukey's multiple comparisons test.

−368 bp, and −359 bp) in the Sost promoter (Fig. 6f). We generated one truncated form of the Tnfsf11 promoter and two truncated forms of the Sost promoter. Deletion mapping identified an AP2α binding site between −1200 and −738 bp of the Tnfsf11 promoter (Fig. 6e) and another binding site between −803 and −333 bp of the Sost promoter (Fig. 6f). Mutating a putative motif at −800 bp of the Tnfsf11 promoter from GCCN3GG to GATN3TA (Fig. 6e) and mutating a putative motif at −368 bp or −359 bp of the Sost promoter from GCCN3GG to GATN3TA confirmed these loci as the AP2α binding sites (Fig. 6f). To determine whether CIITA protein binds to the Tnfsf11 or Sost promoters via AP2α, we knocked down AP2α expression in osteocytes using shAp2α (Fig. 6g) and observed decreased levels of H3K14 acetylation (Fig. 6h), CIITA and H3K14 enrichment in the Tnfsf11 or Sost promoter (Fig. 6i). These results indicate that AP2α recruits CIITA

protein to the Tnfsf11 or Sost promoter and therefore induces histone acetylation in the promoters.

**The clinical relevance of osteocyte CIITA in myeloma.** To elucidate the clinical relevance of osteocyte CIITA in the bone status of patients with myeloma, we collected bone marrow biopsy segments from 14 patients with newly diagnosed myeloma and stained with the anti-CIITA antibody. Histologic examination revealed a robust positive correlation between the H-score of CIITA+ osteocytes and the number of bone lesions in myeloma patients (Fig. 7a). Figure 7b highlights osteocytes buried within bone mineral in two myeloma patients: the H-score of CIITA+ was higher in the osteocytes of patient 1 than in the osteocytes of patient 2. Consistently, we found greater numbers of bone lesions

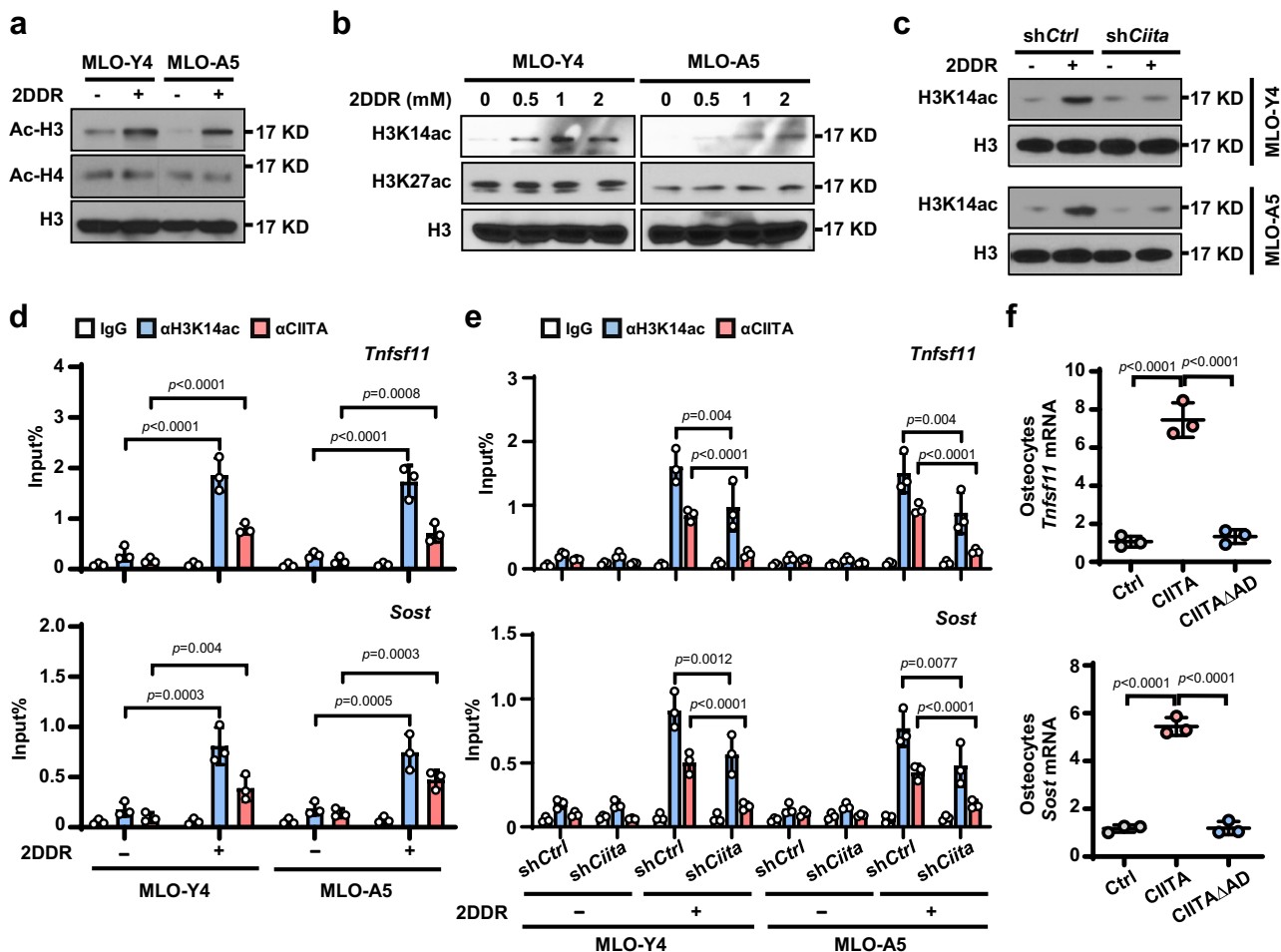

**Fig. 5 CIITA mediates 2DDR-induced histone acetylation at H3K14 in the promoter of *Tnfsf11* and *Sost* genes in osteocytes. a-b** Western blot shows the acetylation levels of total histone H3 (Ac-H3) and H4 (Ac-H4) (**a**) and of histone H3 lysine 14 (H3K14ac) and lysine 27 (H3K27ac) (**b**) in MLO-Y4 or MLO-A5 osteocytes cultured without or with 1 mM 2DDR. H3 levels served as controls. **c** Western blot analysis shows the levels of H3K14ac in sh*Ctrl* or sh*Ciita* MLO-Y4 or MLO-A5 cells cultured without or with 1 mM 2DDR. H3: loading control. Data shown in **a**-**c** are representative of two independent experiments. **d** ChIP assay shows the enrichment of H3K14ac or CIITA in the promoter of *Tnfsf11* or *Sost* genes in MLO-Y4 or MLO-A5 cells cultured without or with 1 mM 2DDR. Data are mean ± SD (n = 3 biological replicates). **e** ChIP assay shows the enrichment of H3K14ac and CIITA in the promoter of *Tnfsf11* or *Sost* genes in sh*Ctrl* or sh*Ciita* MLO-Y4 or MLO-A5 cells cultured without or with 1 mM 2DDR. Data are mean ± SD (n = 3 biological replicates). **f** Shown is the relative expression of *Tnfsf11* or *Sost* in primary osteocytes that were transfected with wild-type *CIITA* cDNA or *CIITAΔAD* (dominant-negative mutant that lacks the HAT activation domain). *P* values were determined using one-way ANOVA with Tukey's multiple comparisons test.

in patient 1 than in patient 2 (Fig. 7a). We also found a positive correlation between the H-score of osteocyte CIITA and levels of bone marrow serum RANKL or sclerostin (Fig. 7c). In addition, we examined the association of myeloma cell expression of TP with RANKL or sclerostin levels in myeloma patients and found a positive correlation (Fig.7d). Correspondingly, we observed a positive correlation between TP expression in myeloma cells and CIITA expression in osteocytes (Fig. 7e).

## Discussion

Using patient samples, knockout mice, and in vitro cultures, this study demonstrates that osteocyte-expressed CIITA promotes bone destruction in multiple myeloma (Fig. 7f). In osteocytes, CIITA binds to the *TNFSF11* or *SOST* promoter, activates histone acetylation at H3K14 in the cytokine gene promoter, and thereby upregulates the expression of RANKL and sclerostin. As a result, the increased cytokines activate osteoclastogenesis and bone resorption and suppress osteoblastogenesis and bone formation. The combination of increased resorption and decreased

formation results in bone destruction. We also demonstrate that myeloma cells upregulate the expression of CIITA in osteocytes through TP/2DDR-mediated STAT1/IRF1 signaling. Our work broadens the understanding of the pathogenesis of multiple myeloma bone disease and indicates that specifically targeting the myeloma-osteocyte interaction can potentially provide therapeutic strategies for this disease.

Osteocytes are the main source of RANKL and sclerostin[4] and thereby drive the process of bone remodeling. RANKL plays a key role in osteoclast activation, and sclerostin can modulate both bone formation and resorption[4,9]. Circulating serum RANKL and sclerostin are significantly elevated in patients with multiple myeloma and are correlated with the status of lytic lesions in patients[18,19]. Attempts to therapeutically target RANKL and sclerostin in myeloma patients have been used to treat bone destruction[20,21]. For example, the anti-resorptive agent denosumab (a monoclonal antibody against RANKL) has been approved by the U.S. Food and Drug Administration for the treatment of osteoporosis, hypercalcemia, and bone destruction in myeloma patients[20]. Administration of neutralizing monoclonal antibodies

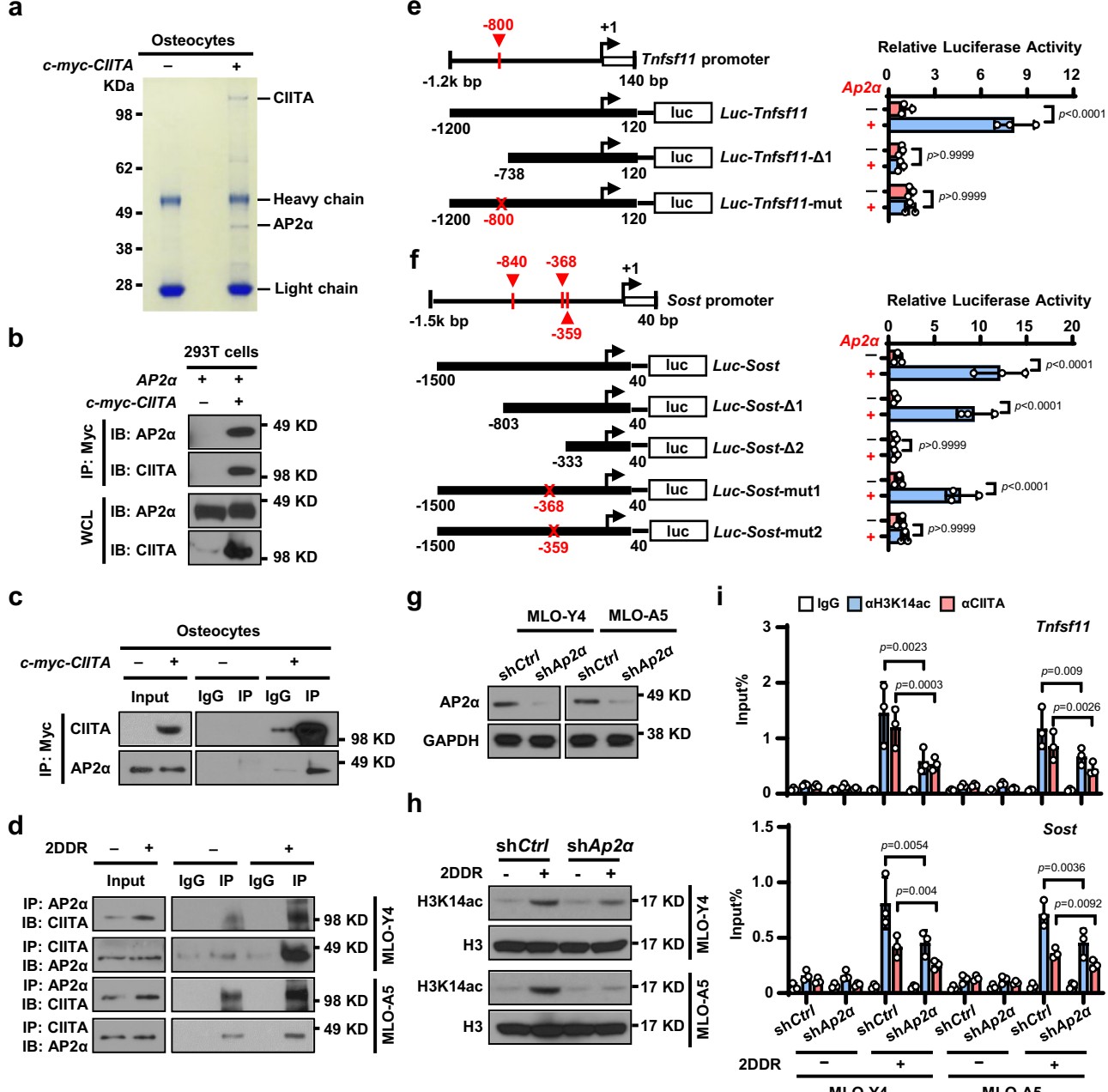

**Fig. 6 AP2α bridges the interaction between CIITA and the promoters of *TNFSF11* and *SOST* in osteocytes. a** Immunoprecipitates pulled down by c-myc-CIITA in osteocytes transfected with a *c-myc–CIITA* plasmid using Coomassie blue staining. **b** Pull-down of AP2α with c-myc-CIITA in HEK293T cells. **c** Co-immunoprecipitation of CIITA with AP2α in osteocytes transfected with *c-myc-CIITA* plasmid. **d** Co-immunoprecipitation of CIITA with AP2α in MLO-Y4 and MLO-A5 osteocytes cultured without or with 1 mM 2DDR. **e, f** Schematic of the *Tnfsf11* (**e**) and *Sost* (**f**) promoter luciferase reporter. Solid boxes: promoter region; red crosses: mutation sites. The luciferase activity of *Luc-Tnfsf11* or *Luc-Sost* constructs was set at 1. ns, not significant. **g-h** Western blot shows the expression of AP2α (**g**) and H3K14ac (**h**) in MLO-Y4 or MLO-A5 cells transfected with non-targeted shRNA (sh*Ctrl*) or *Ap2α* shRNA (sh*Ap2α*) cultured without or with 1 mM 2DDR. (GAPDH and H3: protein loading controls). **i** ChIP assay shows the enrichment of H3K14ac and CIITA in the promoter of *Tnfsf11* or *Sost* genes in MLO-Y4 and MLO-A5 cells expressing sh*Ctrl* or sh*Ap2α* cultured without or with 1 mM 2DDR. Data in (**e**), (**f**), and (**i**) are presented as mean ± SD, n = 3 biological replicates. *P* values were determined using one-way ANOVA with Tukey's multiple comparisons test. Data shown in (**a–d**), (**g**), and (**h**) are representative of two independent experiments.

against sclerostin increases bone mass and decreases osteolytic bone lesion in both animal models and humans[21,22]. But the mechanisms by which myeloma cells regulate RANKL and sclerostin secretion from osteocytes have remained obscure. Other studies demonstrated that myeloma cells trigger apoptosis and higher expression of RANKL in osteocytes, followed by increased osteoclast activity and development of bone lesions[5,23].

Our study elucidates a molecular mechanism for the osteocyte-myeloma interaction, in which myeloma cell TP/2DDR upregulates CIITA expression in osteocytes; the increased CIITA enhances histone acetylation of *TNFSF11* and *SOST* genes; and promotes RANKL and sclerostin secretion from osteocytes. This leads to enhanced osteoclast-mediated bone resorption and decreased osteoblast-mediated bone formation. Our previous

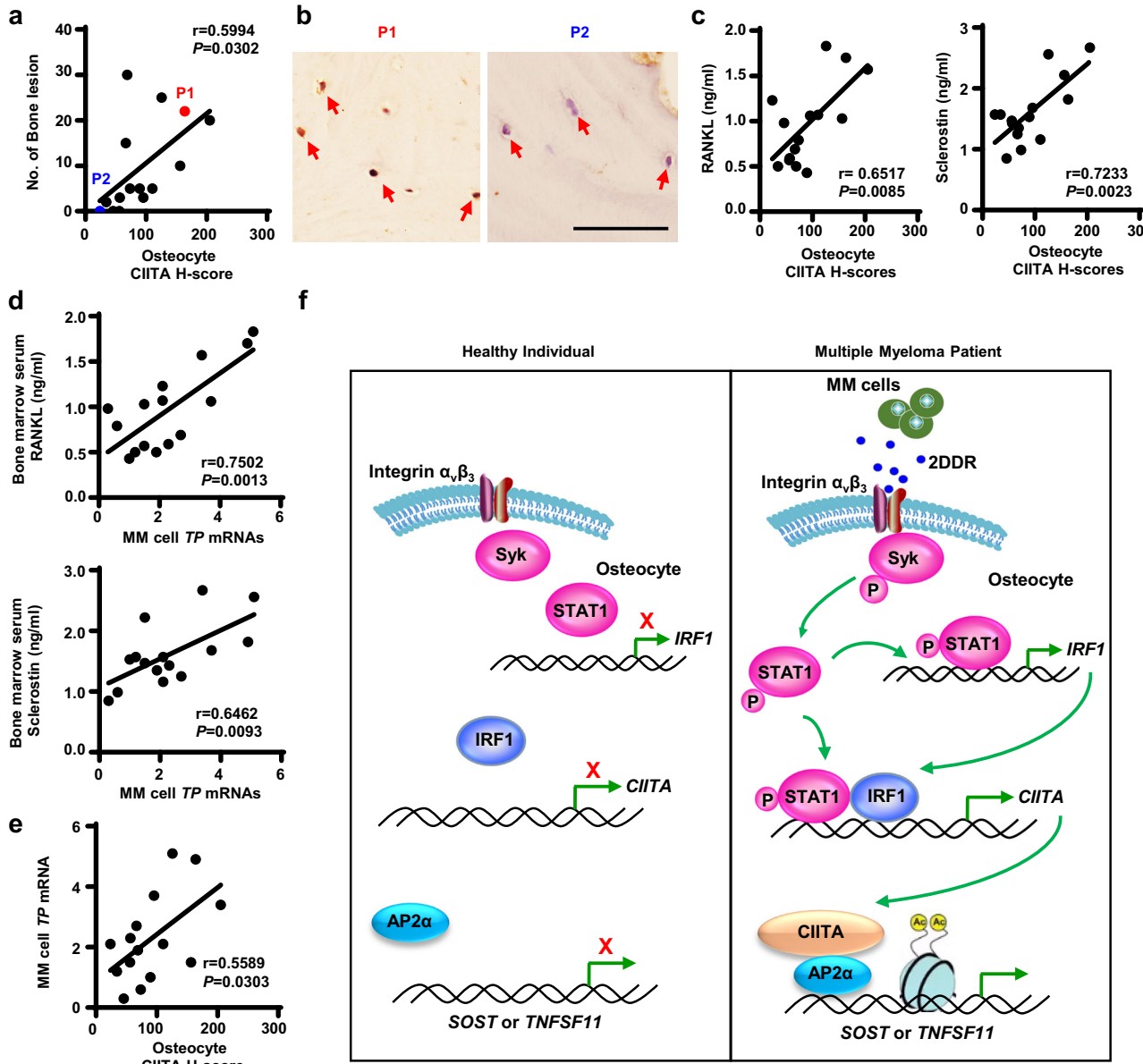

**Fig. 7 Expression levels of CIITA in osteocytes are associated with the status of bone lesions in myeloma patients.** Bone marrow biopsies were randomly collected from myeloma patients ($n = 15$) and then stained with the anti-CIITA antibody. The immunohistochemically stained slides were semi-quantified by H-score, which reflects the totality of percentage of stained cells in each intensity. **a** Correlation between the H-scores of CIITA+ osteocytes and the numbers of bone lesions in myeloma patients. **b** Representative images of immunohistochemical staining show the expression of CIITA in osteocytes in the biopsy segments of the two patients highlighted with red or blue in (**a**). Scale bar, 50 μm. **c**–**e** Bone marrow aspirates of myeloma patients were subjected to ELISA. Shown are the correlation coefficients of the H-scores of CIITA+ osteocytes with bone marrow serum levels of RANKL (**c**, left) or sclerostin (**c**, right), the correlation coefficients of the levels of *TP* mRNA in myeloma cells with bone marrow serum levels of RANKL (**d**, left) or sclerostin (**d**, right), and the correlation coefficients of the H-scores of CIITA+ osteocytes with the levels of *TP* mRNA in myeloma cells (**e**). The correlations were evaluated using Pearson coefficient with two-tailed *P* value. r, correlation coefficient. **f** Depiction of osteocyte CIITA–mediated bone destruction in multiple myeloma. Myeloma TP/2DDR upregulates CIITA expression in osteocytes through the STAT1/IRF1 signaling pathway. The increased CIITA via AP2α as a link binds to the promoter of *Tnfsf11* or *Sost* and activates histone acetylation at H3K14, leading to the expression of RANKL or sclerostin in osteocytes.

study demonstrated that myeloma-secreted 2DDR binds to the receptors on bone cell progenitors, upregulates the methylation of transcriptional factors, and thus regulates osteoblastogenesis and osteoclastogenesis[7]. The combined results from our earlier and current studies indicate that myeloma TP/2DDR regulates bone remodeling through the complex pathways outlined above and lead to mixed bone histology of increased resorption and decreased formation observed in most patients with this disease.

CIITA is a non–DNA binding transcriptional co-activator. CIITA protein contains an acidic transcriptional activation domain, four leucine-rich repeats, and a GTP-binding domain. In the nucleus, CIITA can regulate the transcription of MHC class II genes by interacting with CREB, NF-Y, and the RFX transactivator associated with a highly conserved and unique motif (W/S, X1, X2, and Y) in the promoter region[24,25]. CIITA can also regulate, through histone acetylation, the expression of non-MHC genes that do not possess this motif in their promoter region[26],

suggesting that additional mechanisms must be involved. We have explored the mechanism of how CIITA mediates 2DDR effects on osteocyte cytokine expression. By mutating HAT active domain in CIITA, we confirmed CIITA's HAT activity in osteocytes. By knocking down CIITA expression in osteocytes, we demonstrate that the histone acetylation in cytokine gene promoters induced by 2DDR is dependent upon CIITA. Our study also points to the involvement of AP2α, a transcription factor of AP-2 family. AP-2 factors bind to the consensus sequence 5′-GCCNNNGGC-3′ and play an important biological function in limb and neural tube development[27,28]. The present study demonstrates a function of AP2α as a bridge between CIITA and the promoter of TNFSF11 or SOST. The interaction of CIITA with AP2α and the recruitment of the CIITA/AP2α complex to AP2α-targeted genes may be a general mechanism of regulation of non-MHC genes by CIITA. Whether there are additional components besides AP2α in CIITA/AP2α complex or other HATs involved in the 2DDR-mediated histone activation still needs further investigation.

Previous investigators identified a role and mechanism for CIITA effects in skeletal homeostasis[12]. They found that healthy mice with myeloid-specific CIITA overexpression develop severe osteoporosis caused by enhanced osteoclast differentiation and bone resorption without any change in osteoblast function[12]. Our study differs in that it focuses on the impact of osteocyte-specific CIITA on myeloma-associated bone lesions. Within the tumor microenvironment, osteocytes interact with myeloma cells to promote the development of bone lesions through a complex pathway involving TP, 2DDR and CIITA. Osteocyte-expressed CIITA, distinct from that expressed by myeloid cells, functions to activate osteoclast-mediated bone resorption and also inhibit osteoblast-mediated bone formation. We posit that inhibition of osteoblast activity and reduced bone formation observed in myeloma may be due to the differences in downstream CIITA-mediated signaling pathways. In myeloid cells, CIITA over-expression activates the kinase phosphorylation in the colony-stimulating factor-1 receptor (c-fms) and RANKL-mediated signaling pathways[12]. In osteocytes, CIITA upregulates the transcription of cytokines through acetylation in the cytokine promoter. In multiple myeloma, one may postulate that tumor cell-secreted 2DDR upregulates CIITA expression in myeloid cells and increases osteoclast differentiation through activation of the c-fms/RANKL signaling. A logical next step to clarify this point may be to assess how the TP/2DDR/CIITA axis is involved in myeloma bone disease in an osteocyte-independent manner.

To add further to the complexity, bone remodeling is modulated not only by bone cells but also by immune cells; the bone-immune interaction is involved in the pathogenesis of benign bone diseases such as postmenopausal osteoporosis and rheumatic diseases[29–31]. CIITA, known to activate MHC-II expression in immune cells, has been shown to have an indirect effect on bone[32]. For example, CIITA-induced T cell activation contributes to ovariectomy-induced bone loss[32]. Within the myelomatous bone microenvironment, it may be reasonable to speculate that the TP/2DDR/CIITA axis drives the immune MHC-II antigen presentation machinery leading to bone loss. These possibilities, if proven, would expand TP/2DDR/CIITA as a common mechanism in tumor-associated bone disease.

Collectively, our results demonstrate the mechanism by which myeloma cells upregulate osteocyte derived RANKL and sclerostin in myeloma. We also describe an unexplored mechanism in which CIITA regulates non-MHC class II genes. Implicit in these results is that targeting the integrin $\alpha_v$/STAT1/CIITA signaling pathways could be a potential strategy for the treatment of myeloma-induced bone disease. Finally, our findings may have broader implications for the genesis of osteolytic lesions in other solid malignancies, which often metastasize to bone and create lytic lesions.

## Methods

**Cell lines and primary cells**. The human myeloma cell line ARP-1 was established and provided by the University of Arkansas for Medical Sciences. Murine myeloma Vk*MYC cell line (Vk12598) was established and provided by the Mayo Clinic. HEK293T (CRL-3216), Raw264.7 (TIB-71), and MC3T3-E1 (CRL-2593) cells were purchased from the American Type Culture Collection. MLO-Y4 (EKC002) and MLO-A5 (EKC003) cells were purchased from Kerafast, Inc. Primary myeloma cells were isolated from bone marrow aspirates from patients with myeloma using anti-CD138 antibody-coated magnetic beads (130-051-301, Miltenyi Biotec). Myeloma cells were maintained in RPMI 1640 medium with 10% fetal bovine serum, and HEK293T cells were cultured in DMEM medium with 10% FBS.

Osteocytes, osteoblasts, monocytes, and MSCs were isolated from human or mouse bone pieces[8,33–35]. For isolation of osteocytes and osteoblasts, bone marrow cells were removed, and bone pieces were washed three times in PBS and dissected into small pieces. The bone pieces were incubated at 37 °C for 20 min with shaking in 2 mg/ml collagenase type II (Sigma-Aldrich) and washed with 1× PBS. This process was repeated for three times. To extract osteoblasts, the bone pieces were cultured with Dulbecco's modified Eagle medium (DMEM). For osteocyte isolation, the bone pieces were further digested with a cyclical 5 mM EDTA and 0.1% BSA in 1× PBS and collagenase type II for at least three times. The digested solution was centrifuged, and cell pellets were collected. Osteocytes were maintained in α-Minimum Essential Medium with 5% FBS and 5% fetal calf serum.

To collect osteocyte CM, osteocytes were cultured alone or co-cultured with myeloma cells at the bottom of the well, where myeloma cells were seeded in the insert of the transwell. After two days, transwell with myeloma cells were removed. Osteocytes were cultured in fresh medium for another two days, and the supernatants were collected as CM.

Patient samples were provided by the Myeloma Tissue Bank at The University of Texas MD Anderson Cancer Center or Xiamen University. Informed consent was obtained from study participant and collection of patient samples was conducted by both institutions in accordance with the criteria set by the Declaration of Helsinki. Bone marrow samples were obtained mostly from the iliac crest of patients with myeloma during standard procedure for clinical diagnosis, and the residual samples were used for research. Bone lesions in the study participants were characterized by radiologists who were blinded to our study. The investigators who performed the bench studies were blinded to the bone lesion status of the patients. This study was approved by the MD Anderson Institutional Review Board (PA12-0034) and the Ethics Committee of Xiamen University.

**Plasmids and reagents**. The plasmids myc-CIITA, AP2α, and Ap2α were purchased from GeneCopoeia. Except where specified otherwise, all chemicals were purchased from Sigma-Aldrich; all neutralizing antibodies and enzyme-linked immunosorbent assay (ELISA) kits were purchased from R&D Systems. shRNAs were purchased from Sigma-Aldrich. Human wild-type CIITA (forward: 5′-CA TTCTAGAATGGAGTTGG GGCCCCTAGA-3′; reverse: 5′-TAATCGGCCGG GGTCACCTCCGTATACCCT-3′) or CIITAΔAD (forward: 5′-CATTCTAGAT GCCTGCCAC TGCCTGCGCT; reverse: TAATCGGCCGGGGTCACCTCC GTATACCCT) were sub-cloned into a pCDH-CMV vector.

**In vitro osteoblast and osteoclast formation and function assays**. Mature osteoblasts were generated from MC3T3-E1 with osteoblast medium[36]. Bone formation activity of osteoblasts was determined using Alizarin red-S (Sigma-Aldrich) staining[36]. MLO-A5 was cultured in osteoblast medium for 14 days to obtain mature osteocytes before use.

Raw264.7 cells were cultured to obtain the mature osteoclasts[37]. Briefly, Raw264.7 cells were cultured with or without a low dose of RANKL (10 ng/ml) or osteocyte CM for 7 days to induce mature osteoclast formation. TRAP staining for the detection of mature osteoclasts was performed using a leukocyte acid phosphatase kit (Sigma-Aldrich) according to the manufacturer's instructions. The number of TRAP+ cells per mm$^2$ were counted.

**Western blot analysis**. Cells were harvested and lysed with 1 × lysis buffer (Cell Signaling Technology, CST). Cell lysates were subjected to SDS-PAGE, transferred to a polyvinylidene difluoride (PVDF) membrane and immunoblotted. Uncropped blots and gels are provided in Source Data file.

Antibodies used in the immunoblotting are as following: CIITA (3793, CST, 1:1000), GAPDH (5174, CST, 1:5000), Phospho-p44/42 MAPK (Erk1/2) (Thr202/ Tyr204) (4370, CST, 1:1000), p44/42 MAPK (Erk1/2) (4695, CST, 1:2000), Phospho-Akt (Ser473) (4058, CST, 1:1000), Akt (9272, CST, 1:2000), Phospho-Syk (Tyr525/526) (2710, CST, 1:1000), Syk (2712, CST, 1:1000), Phospho-STAT1 (Tyr701) (9167, CST, 1:1000), STAT1 (14994, CST, 1:1000), Phospho-JAK1 (Tyr525/526) (74129, CST, 1:1000), JAK1 (3332, CST, 1:1000), Histone H3 (4499, CST, 1:5000), IRF1 (8478, CST, 1:1000), Histone H3 (acetyl K9 + K14 + K18 + K23 + K27) (ab47915, abcam, 1:1000), Histone H4 (acetyl K5 + K8 + K12 + K16) (clone: EPR16606, ab177790, abcam, 1:1000), Acetyl-Histone H3 (Lys14) (D4B9) (7627, CST Technology, 1:1000), Acetyl-Histone H3

(Lys27) (D5E4) (8173, CST, 1:1000), Rabbit (DA1E) mAb IgG Isotype Control (3900, CST, 1:100), and AP-2α (C83E10) (3215, CST, 1:1000).

**Quantitative real-time PCR**. Total RNA was isolated using a RNeasy kit (QIAGEN). An aliquot of 1 μg of total RNA was subjected to reverse transcription with a SuperScript II reverse transcription PCR kit (Invitrogen) according to the manufacturer's instructions. Quantitative PCR was performed using SYBR Green Master Mix (Life Technologies) with the QuantStudio 3 Real-Time PCR System (Life Technologies). The primers used are listed in Supplemental Table 2.

**Luciferase assay in vitro**. Vectors with *Tnfsf11* and *Sost* promoter region were constructed as follows: full-length forms (*Luc-Tnfsf11*: −1.2 kb to 120 bp; *Luc-Sost*: −1.5 kb to 40 bp), truncated forms (*Luc-Tnfsf11-Δ1*: −738 bp to 120 bp; *Luc-Sost-Δ1*: −803 bp to 40 bp; *Luc-Sost-Δ2*: −333 bp to 40 bp), and mutated forms (*Luc-Tnfsf11*-mut: GCCN₃GG to GATN₃TA; *Luc-Sost*-mut1 and *Luc-Sost*-mut2: GCCN₃GG to GATN₃TA) were subcloned into a pGL2 vector, and their transcriptional activities in HEK293T cells were examined using a Dual-Luciferase Reporter Assay System (Promega) according to the manufacturer's instructions. The luciferase activity of full-length *Luc-Tnfsf11* and *Luc-Sost* constructs was set at 1. The primers used in the subcloning are listed in Supplemental Table 3.

**Cell proliferation assays, flow cytometry, and ELISA**. The proliferation of osteocytes was assessed using a CellTiter 96 AQueous One Solution Cell Proliferation Assay (Promega). The cells were fixed and stained with antibodies against the surface markers CD44, CD105. Some fixed cells were permeabilized and then stained with antibodies against DMP-1, sclerostin, osteocalcin, and COL1A1. The stained cells were measured by a BD LSRFortessa flow cytometer (BD Biosciences). The results were analyzed using FlowJo software (version 10). In addition, mouse serum was collected and measured using an ELISA kit (R&D Systems) according to the manufacturer's instructions.

Antibodies used in flow cytometry are as following: APC anti-mouse/human CD44 (clone: IM7, 103011, BioLegend, 1:200), APC anti-mouse CD105 (clone: MJ7/18, 120413, BioLegend, 1:200), Mouse DMP-1 (AF4386, R&D Systems, 1:200), Mouse sclerostin (Clone: 248121, MAB1589,R&D Systems, 1:200) Mouse osteocalcin (E6) (sc376835, Santa Cruz, 1:100), Mouse COL1A1 (clone: EPR24331-53, ab270993, abcam, 1:300), Goat anti-Mouse IgG (H + L) Cross-Adsorbed Secondary Antibody, APC (A-865, Thermo Fisher scientific, 1:300), and APC-H7, Mouse IgG1, Isotype Control (560167, BD Biosciences, 1:200)

**Immunofluorescent staining**. Osteocytes were fixed in 4% paraformaldehyde and stained with anti-podoplanin antibody (8.1.1; sc-53533, Santa Cruz Biotechnologies, 1:50), an osteocyte marker, followed by staining of Alexa Fluor 488-labeled goat anti-hamster IgG (H + L) Cross-Adsorbed Secondary Antibody (A-21110, Thermo Fisher, 10 μg/mL)[34]. DAPI was used to stain nuclei. Osteocyte images were acquired using light and fluorescent microscopes.

**Immunohistochemistry**. Formalin-fixed, paraffin-embedded sections of bone marrow biopsy samples obtained from patients with myeloma were deparaffinized and stained[38]. Slides were stained with anti-CIITA antibody (LifeSpan BioSciences) using an EnVision System (Dako) following the manufacturer's instructions and counterstained with hematoxylin. The immunohistochemically stained slides were semi-quantified by H-score. A formulation of H-score = ΣPi(i + 1) was used to calculate the value, where "Pi" is the percentage of stained cells in each intensity category (0–100%) and "i" is the intensity indicating weak (i = 1), distinct (i = 2) or very strong (i = 3)[39].

**Immunoprecipitation and pull-down assays**. Cells were lysed and incubated on ice for 15 min. The total protein lysate was immunoprecipitated with an agarose-immobilized antibody at 4 °C overnight. After being washed six times, the beads were spun down and resuspended in SDS loading buffer. Pull-down samples were run on an SDS-PAGE gel along with a 5% input sample and transferred to a PVDF membrane for immunoblotting. IgG was used as a control, and total cell lysates were used as input controls. For the pull-down assay, HEK293T cells were transfected with plasmid carrying either *c-myc-CIITA* or AP2α. Lysates of the cells pulled down with c-Myc beads (20169, Thermo Fisher Scientific) were further incubated with cell lysates transfected with *AP2α*. The immunoprecipitates and whole-cell lysates were immunoblotted. Cells not transfected with *myc-CIITA* plasmid or whole-cell lysates served as controls.

**ChIP assay**. Cells were fixed in 4% formaldehyde and sonicated to prepare chromatin fragments. Chromatin samples were immunoprecipitated with antibodies against H3K14ac, CIITA, phosphorylated STAT1, and control IgG at 4 °C for 3 h. Immunoprecipitates and total chromatin inputs were reverse cross-linked; DNA was isolated and analyzed using PCR. The primer sequences used are listed in Supplemental Table 4. Relative fold enrichment was calculated by

determining the immunoprecipitation efficiency (ratio of the amount of immunoprecipitated DNA to that of the input sample).

**Mass spectrometry**. Lysate of human osteocytes transfected with *c-myc-CIITA* was pulled down by anti–c-Myc agarose and subjected to an SDS-PAGE gel. Gel bands stained with Coomassie blue were excised separately, alkylated, and digested with trypsin. The tryptic peptides were then analyzed using a nano-LC/MS/MS (Thermo Fisher Scientific) coupled with an 1100 HPLC (Agilent Technologies). The MS/MS spectra were searched using the SEQUEST software program with the BioWorks Browser (version 3.3.1; Thermo Fisher Scientific) against the NCBI database. Osteocytes transfected with an empty vector served as controls.

**In vivo mouse experiments**. Male or female aged 6–8 weeks of CB.17 SCID or C57BL/6 mice purchased from Harlan Laboratories, *Dmp-1*-cre/ERT2 mice purchased from The Jackson Laboratory, and homozygous *Ciita2*flox/flox mice purchased from Charles River Laboratories, were maintained in American Association for Laboratory Animal Science–accredited facilities. All mice were maintained on a standard light and dark cycle with food and water. The animal room had a controlled temperature (23–24 °C), humidity (60.5%). Cage are cleaned every 3–4 days and supplies of water and food are checked daily. All in vivo mouse studies were approved by the UT MD Anderson Institutional Animal Care and Use Committee. End points for humane euthanasia were signs of pain or discomfort including sluggish movement, diarrhea, rapid development of abdominal distension, and cachexia. These limits were not exceeded.

To generate the Cre-expressing *Ciita*-knockout mice, *Ciita*flox/flox mice were first crossed with transgenic mice expressing Cre recombinase under the control of the *Dmp1* promoter. Progeny with Cre expression (*Dmp1*-cre/ERT2-*Ezh2*flox/+) were then back-crossed with *Ciita*flox/flox mice to generate osteocyte-specific *Ciita*flox/flox knockout mice. The primers used for genotyping were JMF1 (5′-CCTA GGAGCCACGGAGCTG-3′), PIIIF3 (5′-TGCCCAAATAGGAGCATTAC-3′), JMR1 (5′-TCCAGAGTCAGAGGTGGTC-3′), *Cre*/forward (5′-ATTGCTGTCA CTTGGTCG-TGGC-3′), and *Cre*/reverse (5′-GGAAAATGCTTCTGTCCGTTT GC-3′). JMR1 and JMF1 were used to amplify wide-type bands (520 bp), and JMR1 and PIIIF3 were used to amplify knockout bands (1000 bp). Western blotting was conducted to confirm ablation of Ciita expression in the osteocytes of randomly selected mice.

To establish myeloma in vivo, WT or osteocyte-specific *Ciita*-KO male or female mice were intrafemorally injected with the murine myeloma cell line Vk12598 (1 × 10⁶ cells/mouse). The mice injected with equal volume of 1× PBS served as controls. In xenograft myeloma mouse model, human myeloma cells (5 × 10⁵ cells/mouse) were intrafemorally injected into NSG mice. In some experiments, mice were treated with TP inhibitors 7DX (200 μg/kg) or TPI (300 μg/kg) intraperitoneally three times a week. Mice received PBS as vehicle control. For monitoring the tumor burden, serum samples were collected from the mice weekly and tested for the presence of myeloma-secreted M-proteins using ELISA analysis. Mouse femurs were scanned with Scanco micro–computed tomography (μCT) −40 system (Scanco Medical) and analyzed with Scanco (version 6.1) or Microview (version 2.5, Parallax Innovations) software. Bone tissues were fixed in 10% neutral buffered formalin and decalcified, and sections of them were stained with toluidine blue or TRAP following standard protocols. Both analyses were done using the BIOQUANT OSTEO (version 18.2.6) software program (BIOQUANT Image Analysis Corporation).

**Whole-transcriptome sequencing**. For RNA-seq[40], RNAs were isolated from mouse osteocytes and then purified with the TruSeq Stranded Total RNA Sample Preparation Kit (Illumina). RNA-seq libraries were synthesized according to the manufacturer's instructions (Illumina). Pooled libraries were quantified using the KAPA Library Quantification Kit (Kapa Biosystems), examined for size distribution using the Fragment Analyzer (Advanced Analytical), and, using the 76-bp paired-end format, sequenced in four lanes of the Illumina HiSeq 4000 Sequencer.

**Statistical analysis**. Statistical significance was analyzed using the SPSS software program (version 10.0; IBM Corporation) or GraphPad Prism (version 9); with two-tailed unpaired Student *t*-tests for comparison of two groups, and one-way analysis of variance (ANOVA) with Tukey's multiple comparisons test for comparison of more than two groups. P values less than 0.05 were considered statistically significant. Experiments were repeated at least two times and similar results were obtained.

**Reporting summary**. Further information on research design is available in the Nature Research Reporting Summary linked to this article.

## Data availability

All data associated with this study can be found in the paper or the supporting documents. Source data are provided with this paper. Uncropped scans of immunoblots and gels are provided in the Source data. The RNA-seq data generated in this study is available at GEO database (GSE200987).

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

## Acknowledgements

We thank The University of Texas MD Anderson Myeloma Tissue Bank. This research was supported by the National Institutes of Health/National Cancer Institute (R01 awards CA190863 and CA193362, J.Y.), the American Cancer Society (Research Scholar Grant 127337-RSG-15-069-01-TBG, J.Y.), and the Cancer Prevention Research Institute of Texas (Scholar of CPRIT Established Investigator Award RR190108, T.A.G.). Supported also by the NIH/NCI under award number P30CA016672 (Core Labs) and used the Small Animal Imaging Facility, Bone Histomorphometry, and Research Histology Core Laboratory. We would like to thank Sarah Bronson, ELS, Research Medical Library, The University of Texas MD Anderson Cancer Center, who edited the manuscript.

## Author contributions

H.L. and J.Y. designed all experiments and wrote the manuscript; H.L., J.H., R.B-Y., Z.L., R.L., Z.W., D.B., and Y.H. performed experiments and statistical analyses; P.L. provided the myeloma patient samples; T.A.G. and R.F.G. provided critical suggestions. All authors reviewed the final manuscript.

## Competing interests

The authors declare no competing interests.
