## [Peer Review File · Nature Communications]

Reviewers' Comments:

Reviewer #1:

Remarks to the Author:

This is an elegant and comprehensive paper that describes the mechanism by which multiple myeloma (MM) cells induce bone loss. While it has been reported that MM can increase RANKL and Wnt ligand inhibitor levels, here the authors demonstrate that myeloma cell TP/2DDR upregulates CIITA expression in osteocytes, and the increased CIITA enhances histone acetylation of TNFSF11 and SOST genes and promotes RANKL and sclerostin secretion from osteocytes, leading to enhanced osteoclast-mediated bone resorption and decreased osteoblast-mediated bone formation.

Data are well presented, controls are in place and conclusions are based on the results. Authors should, however, provide some more technical information. For example osteocyte FACS gating strategy, and more info about intracellular FACS staining are missing. Furthermore, it is not clear what is the difference in the isolation protocol of OBs versus osteocytes and how the culture medium for OB vs osteocytes was prepared. Authors should also provide images of the OB and osteocyte populations in addition to RT PCR for cell specific markers. Please specific acronyms.

Specific comments:

Why is tumor growth unchanged in CIITA cKO mice? Others have shown that targeting OC-mediated resorption reduces tumor growth in bone. Authors need to exclude any potential pro-tumorigenic OC-independent effects of CIITA cKO osteocytes and repeat the experiment by adding ZA or anti-RANKL and determine tumor growth in this setting.

Does alpha-v blocking Ab prevent in vivo bone loss by abrogating the 2DDR effects of MM cells on the osteocytes?

Most of the signaling experiments are done using cell lines stimulated with 2DDR. Authors should confirm some of the signaling data using primary osteocytes exposed to MM overexpressing TP compared to cells with lower TP.

HDAC inhibitors have been shown to increase osteogenesis in MM. It would be great if authors could determine whether HDAC inhibitor effects on osteogenesis are dependent on CIITA activity on SOST promoter in osteocytes.

Reviewer #2:

Remarks to the Author:

This is a novel and interesting paper that provides mechanistic insights on the role of osteocytes in the bone lesions of multiple myeloma (MM). The study includes in vitro experiments as well as animal and human studies. The approach is comprehensive and state of the art.

My most relevant concern regards figure 2. Micro CT scanning is much superior to bone histomorphometry for the assessment of indices of trabecular volume and structure. Quantitative mCT data (BV/TV, Tb.Th, Tb.N and Tb.Sp) should be shown and conclusions drawn from these data, rather than from mCT images of one bone per group.

Figure 1L shows correlation between CIITA levels and static indices of bone formation. The link between CIITA and bone formation would be strengthened by the addition of dynamic indices of bone formation (MAR and BFR/BS). It is surprising that these data are not shown, considering that BFR/BS is shown for the experiments depicted in figure 2A. MAR and BFR/BS should be shown in both figure 1 and 2.

Minor issues

Lines 96 and 97, fig 1 E. Only a few of the factors shown in Fig 1E are osteoclastogenic cytokines. The upregulated osteoclastogenic factors (e.g. TNF and RANKL) should be specifically mentioned in the results.

Line 122. BV/TV is not an index of bone formation. Please edit this sentence.

CIITA is a known bone regulator. Yet, the osteocytic deletion of CTTA did not alter BV/TV in control (no MM) mice. This appears to argue against a role of osteocytic CTTA in bone remodeling. Can the author explain this apparent discrepancy?

All bar graphs should be replaced by graphs showing individual dots.

Reviewer #3:

Remarks to the Author:

Major claims. The paper by H. Liu and coworkers discloses a previously unrecognized circuit driving myeloma bone disease. In a nutshell, they demonstrate a causal role played by osteocytes, matrix-embedded cells known to form a mechano-sensing network that orchestrates physiologic bone remodelling by bone-forming osteoblasts and bone-resorbing osteoclasts. The paper shows that myeloma cells induce osteocytes to express CIITA (MHC class II transactivator), a non-DNA binding transcriptional regulator essential to antigen presenting cells, which in turn drives expression of the osteolytic cytokines, RANKL and sclerostin, whose secretion accounts for suppressed bone-forming and heightened bone-resorbing activity, respectively by osteoblasts and osteoclasts, thereby mediating bone destruction. The work also describes the mechanisms whereby myeloma cells induce CIITA, i.e. through 2-deoxy-D-ribose (2DDR), produced by thymidine phosphorylase (TP), an axis previously associated with myeloma bone disease by the same Authors, though via direct effects on osteoblast and osteoclast precursors. Here, 2DDR release by myeloma cells triggers osteocyte CIITA expression via Stat1-IRF1 signaling. CIITA then imprints osteolytic cytokine gene expression via histone H3 lysine 14 acetylation at the promoters of genes encoding sclerostin and RANKL. Finally, patient expression data are provided to clinically validate the identified circuit, correlating osteocyte CIITA expression with bone lesions, soluble RANK and sclerostin abundance in bone marrow sera, and TP expression in myeloma cells.

Novelty and trans-disciplinarity. Bone destruction is almost invariably associated with, and central to the fatal evolution of, multiple myeloma, a frequent and still incurable cancer. Osteocytes and osteocyte-produced osteolytic cytokines are already known to play a key role in MBD, as well as myeloma cell-released 2DDR. However, the findings reported in this work are novel in that they provide a comprehensive mechanistic understanding of how myeloma cells instruct osteocytes, and how osteocytes in turn instruct bone cells, disrupting bone homeostasis and causing bone disease. Furthering its relevance, in perspective, the circuit defined may play a role in the skeletal damage induced by other cancers that metastasize to bone.

Weaknesses. The work integrates multiple powerful approaches and is illustrated clearly and accurately. However, a number of concerns and weaknesses emerge that compromise overall rigor and some key messages.

The main concerns relate to specific mechanistic conclusions. Being TP-DDR release by myeloma cells and osteocyte-produced sclerostin/RANKL already known to play a key role in myeloma bone disease, most of the work's novelty relies on accurate mechanistic dissection. However, some conclusions are not conclusively demonstrated. For example, the claimed direct transactivating role of CIITA on osteolytic cytokine genes is based: i) on ChIP experiments that have been shown to generate false positive targets in the absence of CIITA null controls; and ii) on the interaction of CIITA with AP2 alpha, largely based on exogenously expressed tagged CIITA. The Authors' conclusions require more convincing evidence, in view of stringent genome-wide ChIP-chip experiments that proved CIITA an oligotropic transcription factor, i.e., with very few targets, mainly, if not exclusively, dedicated for antigen presentation (Krawczyk et al, PLoS Genetics 2008). Another example, the role of the putative histone acetyl-transferase (HAT) activity of CIITA in the transcriptional activation of osteolytic cytokine genes is given for granted, but is not universally recognized in CIITA-driven direct activation of canonical MHCII-related targets. The effects of CIITA on sclerostin and RANKL-coding genes could in fact be indirect, and H3k14 acetylation merely a marker of their activation, rather than a mechanism.

An additional concern, the results are not easily reconciled with previous work, quoted by the Authors, that implicated myeloid-specific CIITA expression in the regulation of osteoclast

differentiation and physiological bone remodelling (Benasciutti et al, JBMR 2014). In that work, systemic and myeloid-restricted CIITA overexpressing mice developed severe osteoporosis due to excess osteoclast differentiation and resorption, but osteoblasts and bone formation were unaffected. If osteocyte CIITA suppresses osteoblasts via sclerostin, why did osteoblasts not display defects in bone formation in mice overexpressing CIITA systemically (thus presumably also in osteocytes)? And if CIITA plays a cell-autonomous pro-osteoclastogenic role in osteoclast precursors, macrophage-related cells in which CIITA can be induced, does 2DDR also induce CIITA in osteoclast precursors? And could this account, at least in part, for the observed stimulation of bone resorption in myeloma-bearing mice?

Moreover, the clinical relevance of the identified axis in human disease is suggested based on apparently elegant and insightful studies on primary myeloma samples; however, the data raise few concerns that warrant additional efforts.

In conclusion, the paper fails to fully convince that osteocyte CIITA is the key driver of myeloma bone disease through the mechanisms described in this manuscript. A number of compulsory issues, including those stemming from the above concerns, are detailed below. These should be dealt with by the Authors to achieve conclusive evidence and substantiate, or correct, their experimental conclusions accordingly.

Concerns and compulsory issues.

1. As shown in Figure 2, myeloma-bearing mice lacking CIITA in osteocytes do not develop bone damage, in spite of a tumor burden comparable to control mice that develop bone disease, as shown by similar paraproteinemia. In view of the benefit that myeloma cells are thought to gain from spreading in the bone marrow, how can tumors grow equally well if the bone-invading mechanism is inactive? The Authors should control if there are equal plasma cell counts in bone, despite no bone destruction, and if myeloma cells grow elsewhere, i.e., whether skeletal hospitality is reduced if osteocytes cannot be induced to express osteolytic cytokines. Along the same line, would osteocyte CIITA null recipients show any resistance to myeloma growth at later time points?
2. The successful use of reductionist cell systems, such as osteoblast and osteoclast precursor cell lines, is a plus, but the osteoclastogenic assays with Raw 264.7 cells shown in Figs 1, 2 and 4 yield very scarce multinucleated cells, limiting reliability. The Authors should confirm the stimulatory effects of osteocyte-conditioned medium also on osteoclast generation from primary murine bone marrow monocytes, where better yields would also enable to test the expected dose-dependency.
3. Expression of the CIITA gene can be constitutive or inducible, depending on three independent promoters – pI, pIII and pIV – respectively in dendritic cells, B cells and IFN-gamma-induced cells. In the paper, CIITA appears expressed already in basal conditions in primary osteocytes and the entire osteoblast lineage (Fig.1I and 2A). Can the Authors provide formal evidence to exclude any contamination by antigen presenting cells? What CIITA isoform is constitutively expressed? The data also show further induction of CIITA in osteocytes by myeloma cell-conditioned media or 2DDR (Fig.3A,C). The Authors also show enrichment of phospho-STAT1 at “the promoter of Ciita gene” in response to 2DDR. The identity of the specific promoter driving inducible CIITA expression in osteocytes under 2DDR stimulation (and if it is distinct from that mediating basal expression) should be defined.
4. An intrinsic histone acetyl transferase (HAT) activity has been formally recognized in CIITA but has not been universally implicated in its mechanism of action. The data presented in Figure 5 do not prove that CIITA’s HAT mediates the activation of osteolytic cytokine gene promoters, for H3K14 acetylation may solely reflect transcriptional activation, with other HATs involved, and do not necessarily implicate CIITA as the responsible HAT. Targeted mutagenesis experiments are needed to causally implicate CIITA’s HAT activity in osteolytic cytokine gene expression in osteocytes. However, these would not rule out the possibility of an indirect effect of CIITA on cytokine genes (see below issue no.5).
5. CIITA is a unique transcriptional co-activator that, unlike most pleiotropic factors, has been shown to target relatively few genes, all related to MHC-mediated antigen presentation (Krawczyk et al, PLoS Genetics 2008). This work also demonstrated the absolute requirement of CIITA null controls to rule out false-positive genes and identify specific targets. This casts doubts on RANKL and SOST being direct CIITA targets, as concluded based on Figure 5D. The Authors need to include CIITA null cells to ensure that the detected association of CIITA with the promoters of candidate target genes encoding osteolytic cytokines is specific and true.
6. Fig. 6A-C: the discovery of AP2 alpha as an interactor of CIITA accounting for direct

transactivation of RANKL and SOST genes and most confirmative experiments rely on exogenously expressed MYC-tagged CIITA. Why, if the Ab immune-precipitates endogenous CIITA (6D)? Together with the concern on SOST and RAKNL genes being direct CIITA targets (see above issue no.5), this weakens further the reconstructed direct recruitment of CIITA on the promoters of the putative target genes.

7. Fig. 6I does not demonstrate that "AP2 alpha recruits CIITA to the *Tnfrsf11* or *Sost* promoter" (as stated at lines 277-278). Rather, it shows that AP2 alpha is needed for 2DDR-induced H3k14 acetylation at those promoters. The Authors must perform ChIP for CIITA protein with the due CIITA null control – as required for Figure 5 (see issue no.5) – to determine, as they intend (line 274), "whether CIITA protein binds to the *Tnfrsf11* or *Sost* promoters via AP2 alpha".

8. Efforts are required to challenge the relevance of the identified mechanism in human disease. Since a minority of patients present without bone disease at diagnosis, correlating CIITA expression with bone lesions is a powerful approach. However, it is difficult to discriminate patients with and without bone disease in Figure 7A. The plot would benefit from removing the patient with ~100 lesions, to spread the distribution and better evaluate the correlation between CIITA expression and bone disease. Similarly, panel B should compare representative patients with no, few and many bone lesions from the main population, excluding P1. What is the "H-score of CIITA+ osteocytes" appearing in Fig.7A, C, and E? It should be briefly explained in the legend and described in the methods section.

9. The model proposed predicts that high CIITA expression in osteocytes is responsible for uncoupling bone remodelling and bone destruction. However, in the cited work that previously implicated myeloid-specific CIITA expression in the regulation of osteoclast differentiation and physiological bone remodelling (Benasciutti et al, JBMR 2014), systemic and myeloid-restricted CIITA overexpressing mice developed severe osteoporosis due to excess osteoclast differentiation and resorption, but bone formation was unaffected. If in CIITA overexpressing mice osteocytes overexpressed CIITA, why did they not display defects in bone formation?

10. In that same paper, CIITA overexpression in the myeloid lineage was shown to increase osteoclastogenesis by cell-autonomously increasing RANKL-induced differentiation. In macrophages, CIITA is inducible. Does the TP-2DDR axis induce CIITA also in osteoclast precursors? The Authors should test this possibility in their in vitro and in vivo systems. If so, MM cell-released 2DDR may cell-autonomously boost osteoclastogenesis, an osteocyte-independent effect that could at least contribute to bone damage.

11. In view of the prime function of CIITA in antigen presentation, the data raise the question as to whether the described TP/2DDR/CIITA axis also drives the expression of the antigen presentation machinery (MHC-II and related molecules), in osteocytes or other cells in the bone marrow of myeloma-bearing mice in vivo. Although MHC-II deficient mice have no overt skeletal defects (Benasciutti et al, JBMR 2014), its upregulation in disease may mediate bone loss. CIITA is known to be necessary and sufficient for MHC-II expression. Moreover, although this is beyond the scope of this paper, should myeloma-induced CIITA expression in osteocytes fail to drive MHC-II expression, this would offer a formidable model to further the understanding of the tissue specific regulation of MHC-II and antigen presentation.

12. The discussion should not refrain from complexity: the previous work by the same Authors already implicated myeloma cell-expressed TP-2DDR in myeloma bone disease via direct effects on bone cells, which most likely coexist with the osteocyte-mediated mechanisms documented herein. This must be clearly acknowledged and discussed.

Minor issues.

1. Lines 72-73: "overexpressing CIITA in mice increases RANKL serum levels and causes severe spontaneous osteoporosis". This statement should be corrected, as the paper cited demonstrates a cell-autonomous stimulatory effect of CIITA within the myeloid-osteoclast lineage, leading to severe bone loss, accounted for by increased responsiveness to RANKL, and not by high RANK serum levels.

2. Figure 1E shows that expression of DKK1, a key osteoblast-suppressing cytokine, is increased in osteocytes from myeloma-bearing mice. Why do the Authors only focus on sclerostin and RANKL thereafter?

3. Figure 1G: along with dramatically up-regulated transcripts including *Ciita*, the volcano plot also shows two distinctly down-regulated genes, whose identity may be interesting to disclose and briefly comment upon.

4. Line 146: "Figure 2D and 2H" should read "Figure 2D-H".

5. As epitomized by figure 2L, the abundance of specific transcripts in samples from myeloma-bearing mice is greatly variable, with only 2 samples justifying the experimental difference and statistical significance, even with a central player like RANKL. Please comment.
6. On Figures S4 and 3: have TP expression and 2DDR release among different human MM lines been correlated? Does differential TP expression correlate with the capacity of selected human MM lines to induce bone loss in mouse recipients?
7. Table S1 appears poorly informative in its present form. All candidates and peptide ratios should be displayed.
8. Line 252, "Because there is no known binding motif of CIITA in the promoter...": this sentence is misleading, for it suggests that CIITA may recognize a specific DNA sequence, whereas it is a non-DNA binding transcriptional coactivator. Please correct.
9. Figure 7: are the samples used to correlate the various signals in patients (osteocyte CIITA expression, bone lesions, BM serum RANK and sclerostin abundance, TP expression in myeloma cells) the same already used in their Sci Transl Med 2016 paper? Is there any overlap of the data in panel D with those already published?

Point-by-point Response [manuscript number: NCOMMS-21-04684-T]

We would like to thank the reviewers for their thoughtful and constructive comments. We believe that with their helpful suggestions, this revised manuscript is substantially improved over the previous submission. All revisions are underlined in the revised manuscript. We hope that you will find our revision satisfactory and thank you very much for your consideration.

Below is point-to-point response to the comments.

Reviewer #1

1. This is an elegant and comprehensive paper that describes the mechanism by which multiple myeloma (MM) cells induce bone loss. While it has been reported that MM can increase RANKL and Wnt ligand inhibitor levels, here the authors demonstrate that myeloma cell TP/2DDR upregulates CIITA expression in osteocytes, and the increased CIITA enhances histone acetylation of TNFSF11 and SOST genes and promotes RANKL and sclerostin secretion from osteocytes, leading to enhanced osteoclast-mediated bone resorption and decreased osteoblast-mediated bone formation.

Data are well presented, controls are in place and conclusions are based on the results. Authors should, however, provide some more technical information. For example osteocyte FACS gating strategy, and more info about intracellular FACS staining are missing. Furthermore, it is not clear what is the difference in the isolation protocol of OBs versus osteocytes and how the culture medium for OB vs osteocytes was prepared. Authors should also provide images of the OB and osteocyte populations in addition to RT PCR for cell specific markers. Please specify acronyms.

Thanks for the advice. We have added more details about intracellular FACS staining in the Method section as suggested. In addition, we have also added the isolation, medium condition for osteocytes and osteoblasts as advised. In supplementary Figure 1, the images showing the morphologies and immunofluorescent staining of osteocyte-specific protein E11/GP38 were also added as advised. After the isolation, we stained the cells with specific markers individually to confirm the characterizations of osteocytes. So, we did not use any special FACS gating strategy.

2. Why is tumor growth unchanged in CIITA cKO mice? Others have shown that targeting OC-mediated resorption reduces tumor growth in bone. Authors need to exclude any potential pro-tumorigenic OC-independent effects of CIITA cKO osteocytes and repeat the experiment by adding ZA or anti-RANKL and determine tumor growth in this setting

We agree with the reviewer that OC-mediated resorption promotes tumor growth in bone. The reason for unchanged tumor growth in CIITA KO mice could be explained by the effects of CIITA on OCs and other types of cells in the bone marrow microenvironment. In this manuscript, we found that CIITA not only promotes OC-induced bone resorption, it can also regulate the differentiation and activity of OBs, which have been shown to affect tumor growth as well. The combined effects from both altered osteoblastogenesis and osteoclastogenesis on tumor growth is in line with previous reports, such as Gooding S, et al., Nature Communications 2019, 10.4533, in which targeting both OCs and OBs did not change MM growth in mice.

3. Does alpha-v blocking Ab prevent in vivo bone loss by abrogating the 2DDR effects of MM cells on the osteocytes?

This is a valid concern, and studies with application of anti- α_v antibody in vivo have been shown to prevent bone loss (Nakamura, I., et al., *J Bone Miner Metab.* 2007; Rodan, S.B. et al, *J Endocrinol.* 1997.). Our previous study has also shown that 2DDR is a ligand for α_v integrin (Liu H et al, *Sci Trans Med*, 2016). Since there are many other ligands that could bind to α_v , we speculate that using neutralizing antibodies against α_v may not be able to pinpoint the unique effect of 2DDR in vivo. We thus conducted additional in vitro studies to address this concern. We cultured osteocytes with 2DDR without or with the antibody, and we found that α_v blocking antibody treatment abrogates 2DDR effects on CIITA, RANKL, and sclerostin expression in osteocytes (Figure 3H).

4. Most of the signaling experiments are done using cell lines stimulated with 2DDR. Authors should confirm some of the signaling data using primary osteocytes exposed to MM overexpressing TP compared to cells with lower TP.

We have provided new data with primary osteocytes as suggested (Figure 3F, 4E, and 5F).

5. HDAC inhibitors have been shown to increase osteogenesis in MM. It would be great if authors could determine whether HDAC inhibitor effects on osteogenesis are dependent on CIITA activity on SOST promoter in osteocytes.

Thanks for the advice. We agree that an HDAC inhibitor can affect SOST-mediated osteogenesis. In this paper, we have focused on the CIITA, which possesses HAT activity that affects histone acetylation. Since the inhibitor is specifically targeting the deacetylation of HDACs, it is not clear whether it could affect the histone acetylation. Whether CIITA is intertwined with the HDAC pathway or it acts in an independent pathway has not been explored, and it is certainly an interesting topic and could be our logical next step.

Reviewer #2

1. This is a novel and interesting paper that provides mechanistic insights on the role of osteocytes in the bone lesions of multiple myeloma (MM). The study includes in vitro experiments as well as animal and human studies. The approach is comprehensive and state of the art.

My most relevant concern regards figure 2. Micro CT scanning is much superior to bone histomorphometry for the assessment of indices of trabecular volume and structure. Quantitative mCT data (BV/TV, Tb.Th, Tb.N and Tb.Sp) should be shown and conclusions drawn from these data, rather than from mCT images of one bone per group.

Thanks for the advice and sorry for the confusion. Figure 2D is indeed the data from quantitative μ CT analysis. In addition, we have provided Tb.Th, Tb.N and Tb.Sp data as advised (Figure 2E-2G). We believe that data from both bone histomorphometry and μ CT give a more complete picture on bone remodeling.

2. Figure 1L shows correlation between CIITA levels and static indices of bone formation. The link between CIITA and bone formation would be strengthened by the addition of dynamic indices of bone formation (MAR and BFR/BS). It is surprising that these data are not shown,

considering that BFR/BS is shown for the experiments depicted in figure 2A. MAR and BFR/BS should be shown in both figure 1 and 2.

Thanks for the advice and we provide the MAR and BFR/BS data in Figure 1L as suggested.

Minor issues

1. Lines 96 and 97, fig 1 E. The upregulated osteoclastogenic factors (e.g. TNF and RANKL) should be specifically mentioned in the results.

Sorry for the confusion. We meant to display the cytokines that osteocytes secrete. We have removed the “osteolytic” from this sentence.

2. Line 122. BV/TV is not an index of bone formation. Please edit this sentence

Thank you for pointing it out. We have revised the sentence.

3. CIITA is a known bone regulator. Yet, the osteocytic deletion of CIITA did not alter BV/TV in control (no MM) mice. This appears to argue against a role of osteocytic CIITA in bone remodeling. Can the author explain this apparent discrepancy?

This is a valid concern. Based on the previous study (Benasciutti et al, JBMR 2014), the effect of CIITA on bone lesion in mice won't be apparent until 24 weeks in age. To avoid the discrepancy in bone density, our experiments were conducted with the mice less than 12 weeks old, where there was no significant difference in BV/TV between wild type and knockout mice.

Reviewer #3

1. As shown in Figure 2, myeloma-bearing mice lacking CIITA in osteocytes do not develop bone damage, in spite of a tumor burden comparable to control mice that develop bone disease, as shown by similar paraproteinemia. In view of the benefit that myeloma cells are thought to gain from spreading in the bone marrow, how can tumors grow equally well if the bone-invading mechanism is inactive? The Authors should control if there are equal plasma cell counts in bone, despite no bone destruction, and if myeloma cells grow elsewhere, i.e., whether skeletal hospitality is reduced if osteocytes cannot be induced to express osteolytic cytokines. Along the same line, would osteocyte CIITA null recipients show any resistance to myeloma growth at later time points?

Our data show clearly that injection of MM cells causes bone lesions, and this is largely (but not completely) reversed in the CIITA osteocyte KO mice. In these experiments there was not a significant difference in tumor volume between WT and KO mice and there was no metastasis of MM cells to other organs in either WT or KO mice (in line with our previous results). We can think of potential explanations for this observation. It is possible that MM cell growth was unaffected because there was still adequate marrow space to expand. There could also be another effect of CIITA that we have not yet identified. We anticipate future studies to explore this effect and the potential effects of CIITA on MM growth. These are complex questions, beyond the scope of this manuscript.

2. The successful use of reductionist cell systems, such as osteoblast and osteoclast precursor cell lines, is a plus, but the osteoclastogenic assays with Raw 264.7 cells shown in Figs 1, 2 and 4 yield very scarce multinucleated cells, limiting reliability. The Authors should confirm the

stimulatory effects of osteocyte-conditioned medium also on osteoclast generation from primary murine bone marrow monocytes, where better yields would also enable to test the expected dose-dependency.

Thanks for the advice. To capture the effect of osteocytes on osteoclastogenesis, we used a low dose of RANKL in our culturing system, therefore the formation of multinucleated OCs was a bit low. Following your advice, we have used primary murine monocytes in lieu of RAW264.7 cells and observed a similar trend.

3. Expression of the CIITA gene can be constitutive or inducible, depending on three independent promoters – pI, pIII and pIV – respectively in dendritic cells, B cells and IFN-gamma-induced cells. In the paper, CIITA appears expressed already in basal conditions in primary osteocytes and the entire osteoblast lineage (Fig.1I and 2A). Can the Authors provide formal evidence to exclude any contamination by antigen presenting cells? What CIITA isoform is constitutively expressed? The data also show further induction of CIITA in osteocytes by myeloma cell-conditioned media or 2DDR (Fig.3A,C). The Authors also show enrichment of phospho-STAT1 at “the promoter of Ciita gene” in response to 2DDR. The identity of the specific promoter driving inducible CIITA expression in osteocytes under 2DDR stimulation (and if it is distinct from that mediating basal expression) should be defined.

This is a valid concern. We are really mindful of potential contamination from other cell types. So we constructed a well-established protocol for isolation of osteocytes, which utilized multiple steps to clean and digest the bone piece before isolation. To exclude the influence from other cell types, we have conducted our experiments using osteocyte-specific knockout mice. We have also used multiple osteocyte cell lines, which are free of antigen presenting cells.

Thanks for the information regarding CIITA various isoforms. In this study, we focused on the role and mechanism of osteocyte CIITA in regulation of bone remodeling when osteocytes were exposed to MM cells. It certainly is interesting to find out whether CIITA expression in osteocytes is constitutive or inducible, which would be a new project to investigate.

4. An intrinsic histone acetyl transferase (HAT) activity has been formally recognized in CIITA but has not been universally implicated in its mechanism of action. The data presented in Figure 5 do not prove that CIITA’s HAT mediates the activation of osteolytic cytokine gene promoters, for H3K14 acetylation may solely reflect transcriptional activation, with other HATs involved, and do not necessarily implicate CIITA as the responsible HAT. Targeted mutagenesis experiments are needed to causally implicate CIITA’s HAT activity in osteolytic cytokine gene expression in osteocytes. However, these would not rule out the possibility of an indirect effect of CIITA on cytokine genes (see below issue no.5).

We were intrigued by your comments and have performed mutation experiments using CIITA Δ AD, the known dominant-negative mutant of CIITA that lacks the HAT activation domain and found that the mutation significantly reduced the expression of RANKL and sclerostin in osteocytes (Figure 5F). This data provides additional confirmation that the HAT activity of CIITA contributes to its effects in osteocytes. We plan future studies to determine whether CIITA is solely responsible to the acetylation in those genes or if there might be other HATs involved in the process. We have addressed this in the discussion section.

5. CIITA is a unique transcriptional co-activator that, unlike most pleiotropic factors, has been shown to target relatively few genes, all related to MHC-mediated antigen presentation (Krawczyk et al, PLoS Genetics 2008). This work also demonstrated the absolute requirement of CIITA null controls to rule out false-positive genes and identify specific targets. This casts doubts on RANKL and SOST being direct CIITA targets, as concluded based on Figure 5D. The Authors need to include CIITA null cells to ensure that the detected association of CIITA with the promoters of candidate target genes encoding osteolytic cytokines is specific and true.

We agree that the most recognized functions of CIITA are focused on the MHC Class II-related genes. The stringent genome-wide ChIP-chip experiments by Krawczyk et al. provided convincing evidence that there are only few targets of CIITA for antigen presentation. This conclusion doesn't conflict with ours, since their studies were conducted using B cells and dendritic cells and are specific for the function of CIITA in immune cells. Other studies, such as one from Benasciutti group (JBMR 2014. 29:290-303), has also implicated a role of CIITA in osteoclastogenesis, which is independent of MHC Class II. In addition, a previous study by Bonewald et al (Endocrinology 2003. 144: 1761) shows that MHC Class II is not expressed in the osteocyte cell line MLO-Y4, suggesting that the effect of CIITA in osteocytes may be independent of the regulation of MHC Class II expression. Therefore, the function and action mechanism of CIITA may vary in different cell types.

In addition, we have updated Figure 5E with the binding of CIITA to cytokine gene promoters in null control cells as advised.

6. Fig. 6A-C: the discovery of AP2 alpha as an interactor of CIITA accounting for direct transactivation of RANKL and SOST genes and most confirmative experiments rely on exogenously expressed MYC-tagged CIITA. Why, if the Ab immune-precipitates endogenous CIITA (6D)? Together with the concern on SOST and RAKNL genes being direct CIITA targets (see above issue no.5), this weakens further the reconstructed direct recruitment of CIITA on the promoters of the putative target genes.

Sorry for the confusion. We have conducted a multi-step approach to examine the interaction between AP2 α and CIITA. Using the pull-down assay, we first examined the potential physical interaction using exogenously transfected HEK293T cells with plasmids expressing c-myc-CIITA or with AP2 α . We mixed the two cell lysates, and pulled down with an anti-c-myc antibody. We detected AP2 α in the mixture, indicating that AP2 α can directly bind to CIITA (Figure 6B). Next, we questioned whether such an interaction can be detected in osteocytes. To address this, we transfected c-myc-CIITA into osteocytes and found that anti-c-myc antibody was able to immunoprecipitate CIITA with endogenous AP2 α from osteocytes (Figure 6C). More importantly, we verified the existence of the complex containing endogenous AP2 α and CIITA in osteocytes by immunoprecipitation with either anti-AP2 α antibodies or anti-CIITA antibodies (Figure 6D). Those experiments suggest a direct interaction between CIITA and AP2 α .

7. Fig. 6I does not demonstrate that "AP2 alpha recruits CIITA to the Tnfrsf11 or Sost promoter" (as stated at lines 277-278). Rather, it shows that AP2 alpha is needed for 2DDR-induced H3k14 acetylation at those promoters. The Authors must perform ChIP for CIITA protein with the due CIITA null control – as required for Figure 5 (see issue no.5) – to determine, as they intend (line 274), "whether CIITA protein binds to the Tnfrsf11 or Sost promoters via AP2 alpha".

Sorry for the confusion. We have updated Figure 6I as advised showing the enrichment of CIITA in cytokine gene promoters in control and AP2 α knockdown osteocytes.

8. Efforts are required to challenge the relevance of the identified mechanism in human disease. Since a minority of patients present without bone disease at diagnosis, correlating CIITA expression with bone lesions is a powerful approach. However, it is difficult to discriminate patients with and without bone disease in Figure 7A. The plot would benefit from removing the patient with ~100 lesions, to spread the distribution and better evaluate the correlation between CIITA expression and bone disease. Similarly, panel B should compare representative patients with no, few and many bone lesions from the main population, excluding P1. What is the “H-score of CIITA+ osteocytes” appearing in Fig.7A, C, and E? It should be briefly explained in the legend and described in the methods section.

In accordance with your advice, we have removed the P1 sample from the group. In addition, we have added more clinical samples to strengthen our data. As a result, we have replaced IHC staining picture with another patient sample in Figure 7B and correlation graphs in Figure 7A, 7C-7E.

9. The model proposed predicts that high CIITA expression in osteocytes is responsible for uncoupling bone remodelling and bone destruction. However, in the cited work that previously implicated myeloid-specific CIITA expression in the regulation of osteoclast differentiation and physiological bone remodelling (Benasciutti et al, JBMR 2014), systemic and myeloid-restricted CIITA overexpressing mice developed severe osteoporosis due to excess osteoclast differentiation and resorption, but bone formation was unaffected. If in CIITA overexpressing mice osteocytes overexpressed CIITA, why did they not display defects in bone formation?

The concern is the transgenic mice from Benasciutti group (JBMR 2014. 29:290-303) and our myeloma-bearing mice both have overexpressed CIITA, but have discrepant bone formation. It is impossible for us to provide a concrete answer for the discrepancy, since there is no head-to-head comparison; and there are many differences between the two systems and the conditions under which the experiments were performed. For example, the Benasciutti group illustrated a mechanism in normal mice with global CIITA overexpression in which osteoporosis developed. We, on the other hand, focused on the tumor microenvironment, where bone cells interact with myeloma cells and formed a vicious cycle to enhance bone lesions. It is also not clear whether the levels of CIITA expression in their systemic or conditional overexpressed mice were comparable with those in our myeloma-bearing mouse model. In this manuscript, we have identified a new mechanism defining an interaction between tumor cells and osteocytes and the function of osteocyte-expressed CIITA in myeloma bone disease.

10. In that same paper, CIITA overexpression in the myeloid lineage was shown to increase osteoclastogenesis by cell-autonomously increasing RANKL-induced differentiation. In macrophages, CIITA is inducible. Does the TP-2DDR axis induce CIITA also in osteoclast precursors? The Authors should test this possibility in their in vitro and in vivo systems. If so, MM cell-released 2DDR may cell-autonomously boost osteoclastogenesis, an osteocyte-independent effect that could at least contribute to bone damage.

Thanks for your suggestion. It is certainly an interesting question and could be a logical next step. For this manuscript, we would have focused on osteocytes, and the role of osteocyte CIITA

in MM-induced bone lesions. As we expand this work in the future the question you raise will be kept clearly in mind.

11. In view of the prime function of CIITA in antigen presentation, the data raise the question as to whether the described TP/2DDR/CIITA axis also drives the expression of the antigen presentation machinery (MHC-II and related molecules), in osteocytes or other cells in the bone marrow of myeloma-bearing mice in vivo. Although MHC-II deficient mice have no overt skeletal defects (Benasciutti et al, JBMR 2014), its upregulation in disease may mediate bone loss. CIITA is known to be necessary and sufficient for MHC-II expression. Moreover, although this is beyond the scope of this paper, should myeloma-induced CIITA expression in osteocytes fail to drive MHC-II expression, this would offer a formidable model to further the understanding of the tissue specific regulation of MHC-II and antigen presentation.

We agree with the reviewer that it would be interesting to find out whether the TP/2DDR/CIITA axis is involved in the antigen presentation machinery. We also agree with the reviewer that this issue is beyond the scope of this study. Nevertheless, it is a great suggestion and appreciated.

12. The discussion should not refrain from complexity: the previous work by the same Authors already implicated myeloma cell-expressed TP-2DDR in myeloma bone disease via direct effects on bone cells, which most likely coexist with the osteocyte-mediated mechanisms documented herein. This must be clearly acknowledged and discussed.

We have revised the Discussion section as advised.

Minor issues.

1. Lines 72-73: “overexpressing CIITA in mice increases RANKL serum levels and causes severe spontaneous osteoporosis”. This statement should be corrected, as the paper cited demonstrates a cell-autonomous stimulatory effect of CIITA within the myeloid-osteoclast lineage, leading to severe bone loss, accounted for by increased responsiveness to RANKL, and not by high RANK serum levels.

Thank you for pointing this out; we have revised as suggested.

2. Figure 1E shows that expression of DKK1, a key osteoblast-suppressing cytokine, is increased in osteocytes from myeloma-bearing mice. Why do the Authors only focus on sclerostin and RANKL thereafter?

We did not focus on DKK1 as it has already been shown to have an important role in myeloma bone disease. However, your point is on target and we plan to include these studies as we expand this work in the future.

3. Figure 1G: along with dramatically up-regulated transcripts including Ciita, the volcano plot also shows two distinctly down-regulated genes, whose identity may be interesting to disclose and briefly comment upon.

The two most down-regulated genes are interferon-activable protein 203 (Ifi203) and myeloid cell nuclear differentiation antigen-like protein (Mndal). There is limited information regarding those genes, and it probably would require a separate project for further investigation.

4. Line 146: “Figure 2D and 2H” should read “Figure 2D-H”

Thanks, we have revised as suggested.

5. As epitomized by figure 2L, the abundance of specific transcripts in samples from myeloma-bearing mice is greatly variable, with only 2 samples justifying the experimental difference and statistical significance, even with a central player like RANKL. Please comment.

Thanks for pointing it out to us. We have done this set of experiment multiple times and obtained the same conclusion. This may not be the best representation, so we have replaced with the results with another independent set of experiment (Figure 2M to 2P).

6. On Figures S4 and 3: have TP expression and 2DDR release among different human MM lines been correlated? Does differential TP expression correlate with the capacity of selected human MM lines to induce bone loss in mouse recipients?

We have investigated these questions in our previous study (Liu H et al, Sci Trans Med, 2016).

7. Table S1 appears poorly informative in its present form. All candidates and peptide ratios should be displayed.

The gel bands were cut out like in Figure 6A and then sent for mass spectrometry analysis. We have revised Table S1 as advised.

8. Line 252, “Because there is no known binding motif of CIITA in the promoter...”: this sentence is misleading, for it suggests that CIITA may recognize a specific DNA sequence, whereas it is a non-DNA binding transcriptional coactivator. Please correct.

Thanks. We have revised as advised.

9. Figure 7: are the samples used to correlate the various signals in patients (osteocyte CIITA expression, bone lesions, BM serum RANK and sclerostin abundance, TP expression in myeloma cells) the same already used in their Sci Transl Med 2016 paper? Is there any overlap of the data in panel D with those already published?

This is a new set of patient samples with no overlap with our previous publication.

Reviewers' Comments:

Reviewer #1:

Remarks to the Author:

The authors addressed my concerns and after the revision the paper is much improved

Reviewer #2:

None

Reviewer #3:

Remarks to the Author:

The Authors have convincingly addressed some of the concerns raised, strengthening their work. However, there are issues that remain unaddressed. These are summarized as follows.

Issue no. 2 - the concern of relying completely on Raw 264.7 cells for osteoclastogenic assays - was only partially addressed, and 2 weaknesses remain. 1) Primary monocytes were added only in one figure panel out of many experiments. As a result, most data depend on an assay - OCgenesis from poorly OCgenic, M-CSF-independent leukemic cells - that is weakly reproducible and inadequately recapitulating normal OCgenesis. 2) Owing to the scarce osteoclast yield in this model, the requested dose-dependency was not tested.

In response to concern no.3, the Authors declined to identify which CIITA isoform is induced by 2DDR in osteocytes, as urged by the current knowledge on the genetic regulation of CIITA expression. Defining the isotype expressed would strengthen the work, offering a framework to understand the basis of constitutive CIITA expression and its induction by myeloma cell-produced 2DDR in osteocytes.

In the context of issue no.8, the Authors haven't defined the H-score in Methods and briefly in the legend to figure 7, as requested.

As requested in concerns 9, 10, and 11, to avoid oversimplification, in the discussion the Authors should briefly acknowledge: 1) the discrepancy of their results with CIITA overexpressing mice not showing depressed OB activity; 2) the unexplored possibility of a myeloid-intrinsic, and osteocyte-independent, role of myeloma-induced CIITA contributing to myeloma bone disease; 3) the possible implication of the identified TP/2DDR/CIITA axis in immune-mediated bone-wasting diseases.

Point-by-point Response [manuscript number: NCOMMS-21-04684A]

We would like to thank the reviewer for the additional thoughtful and constructive comments. We believe that with those helpful suggestions, this revised manuscript is substantially improved over the previous submission. All revisions are underlined in the revised manuscript. We hope that you will find our revision satisfactory and thank you very much for your consideration.

Below is point-to-point response to the comments.

Reviewer #3

The Authors have convincingly addressed some of the concerns raised, strengthening their work. However, there are issues that remain unaddressed. These are summarized as follows.

1. Issue no. 2 - the concern of relying completely on Raw 264.7 cells for osteoclastogenic assays - was only partially addressed, and 2 weaknesses remain. 1) Primary monocytes were added only in one figure panel out of many experiments. As a result, most data depend on an assay – OCgenesis from poorly OCgenic, M-CSF-independent leukemic cells – that is weakly reproducible and inadequately recapitulating normal OCgenesis. 2) Owing to the scarce osteoclast yield in this model, the requested dose-dependency was not tested.

Answer: We have conducted additional experiments using primary monocytes as advised. In combination, we now have Figure 1B, Figure 2N, Figure 4E, Figure 4J, and figure S3E using primary monocytes. We are sorry for the confusion on quantitation of TRAP⁺ cells in our osteoclastogenesis assay. There are two widely accepted methods for the quantitation; the number of TRAP⁺ cells could be presented as either per mm² or per well. We have used the former, and if we convert it into the number of TRAP⁺ cells per well, they are consistent with previous studies (Kim SS, et al, *Molecules* 2022 Jan 14;27(2):501; Wang L, et al, *Cell Signal* 2021 Feb;78:109847; Jin et al., *International Journal of Oral Biology* 2019; 44 (3):89-95). For example, the Figure 2N would look like as panel A below. We have specified our quantitation in the Method section. In addition, we have conducted the dose curve with various concentrations of OS CM, and we could see the upregulation with increased OS CM (panel B) below. Those results confirmed that CIITA expressed by osteocytes promotes osteoclastogenesis.

2. In response to concern no.3, the Authors declined to identify which CIITA isoform is induced by 2DDR in osteocytes, as urged by the current knowledge on the genetic regulation of CIITA expression. Defining the isotype expressed would strengthen the work, offering a framework to understand the basis of constitutive CIITA expression and its induction by myeloma cell-produced 2DDR in osteocytes.

Answer: We have designed a set of new primers targeting isoform I, III, and IV (Benasciutti E, et al., J Bone Miner Res 2014. 29:287-289). Our quantitative PCR assay showed that the isoform IV is the dominate form in osteocytes and can be induced by 2DDR. The data has been added to the manuscript (figure S5).

3. In the context of issue no.8, the Authors haven't defined the H-score in Methods and briefly in the legend to figure 7, as requested.

Answer: We have added the H-score methods in the Methods section and in Figure 7.

4. As requested in concerns 9, 10, and 11, to avoid oversimplification, in the discussion the Authors should briefly acknowledge: 1) the discrepancy of their results with CIITA overexpressing mice not showing depressed OB activity; 2) the unexplored possibility of a myeloid-intrinsic, and osteocyte-independent, role of myeloma-induced CIITA contributing to myeloma bone disease; 3) the possible implication of the identified TP/2DDR/CIITA axis in immune-mediated bone-wasting diseases.

Answer: Thanks for the advice. We have acknowledged the above points in the discussion section as advised.

Reviewers' Comments:

Reviewer #3:

Remarks to the Author:

The Authors have finally addressed the remaining concerns, reaching conclusive evidence for related key issues, clarifying several experimental issues, and discussing the requested aspects.